# Bivalent individualization during chromosome territory formation in *Drosophila* spermatocytes by controlled condensin II protein activity and additional force generators

**Luisa Vernizzi**, **Christian F. Lehner***

Department of Molecular Life Science (DMLS), University of Zurich, Zurich, Switzerland

* christian.lehner@imls.uzh.ch

**Data Availability Statement:** All relevant data are within the manuscript and its Supporting Information files.

## Abstract

Reduction of genome ploidy from diploid to haploid necessitates stable pairing of homologous chromosomes into bivalents before the start of the first meiotic division. Importantly, this chromosome pairing must avoid interlocking of non-homologous chromosomes. In spermatocytes of *Drosophila melanogaster*, where homolog pairing does not involve synaptonemal complex formation and crossovers, associations between non-homologous chromosomes are broken up by chromosome territory formation in early spermatocytes. Extensive non-homologous associations arise from the coalescence of the large blocks of pericentromeric heterochromatin into a chromocenter and from centromere clustering. Nevertheless, during territory formation, bivalents are moved apart into spatially separate subnuclear regions. The condensin II subunits, Cap-D3 and Cap-H2, have been implicated, but the remarkable separation of bivalents during interphase might require more than just condensin II. For further characterization of this process, we have applied time-lapse imaging using fluorescent markers of centromeres, telomeres and DNA satellites in pericentromeric heterochromatin. We describe the dynamics of the disruption of centromere clusters and the chromocenter in normal spermatocytes. Mutations in *Cap-D3* and *Cap-H2* abolish chromocenter disruption, resulting in excessive chromosome missegregation during M I. Chromocenter persistence in the mutants is not mediated by the special system, which conjoins homologs in compensation for the absence of crossovers in *Drosophila* spermatocytes. However, overexpression of Cap-H2 precluded conjunction between autosomal homologs, resulting in random segregation of univalents. Interestingly, *Cap-D3* and *Cap-H2* mutant spermatocytes displayed conspicuous stretching of the chromocenter, as well as occasional chromocenter disruption, suggesting that territory formation might involve forces unrelated to condensin II. While the molecular basis of these forces remains to be clarified, they are not destroyed by inhibitors of F actin and microtubules. Our results indicate that condensin II activity promotes chromosome territory formation in co-operation with additional force generators and that careful co-ordination with alternative homolog conjunction is crucial.

**Funding:** The research was supported by funds obtained from the Swiss National Science Foundation (www.snf.ch), grant number 31003A_179433 (CFL). The funders had no role in study design, data collection and analysis, decision to publish, or preparation of the manuscript.

**Competing interests:** The authors have declared that no competing interests exist.

## Author summary

Fusion of an oocyte with a sperm generates the first cell of a next generation. The resulting diploid state, with a maternally and a paternally derived copy of each chromosome, is maintained during cell proliferation by mitotic divisions. Eventually, a distinct division process, meiosis, is required for oocyte and sperm production. Meiosis generates haploid cells with only one copy of each chromosome. To this end, the two copies are first physically linked into a bivalent chromosome, followed by integration into a bipolar division spindle, which eventually pulls the two copies apart into distinct daughter cells. How do the two chromosome copies find each other? How are inappropriate associations avoided? Distinct strategies have evolved. Recombination and synapsis, which are usually essential for meiosis, are not deployed in *Drosophila* spermatocytes. Moreover, inappropriate associations are prevented by a special process known as chromosome territory formation. Here, we apply time-lapse imaging to study this process and evaluate the role of F actin, microtubules and condensin II. Cytoskeletal dynamics do not appear to contribute. Controlled condensin II activity, while crucial, appears to have support from additional force generators to achieve the extensive separation of bivalents into distinct subnuclear regions.

## Introduction

Meiosis reduces genome ploidy from diploid to haploid with two consecutive divisions, meiosis I (M I) and meiosis II (M II). Successful ploidy reduction depends on stable pairing of homologous chromosomes into bivalents before the onset of M I. The mechanisms that achieve homolog pairing are still poorly understood and they can vary considerably between organisms [1]. In a range of species, including budding and fission yeast, maize, *Caenorhabditis elegans*, mouse and *Drosophila melanogaster* oocytes, centromeric or telomeric regions make initial inter-chromosomal contacts [2]. The early contacting chromosomal regions are often tethered to the nuclear envelope (NE) by meiosis-specific linkage to LINC complexes. Moreover, motor proteins attached to the cytoplasmic side of the LINC complexes drag the tethered chromosomal regions along by moving on F actin or microtubules [2–4]. These rapid prophase movements (RPMs) have been proposed to overcome the rather limited passive diffusion of meiotic chromosomes and increase homolog contact probability. Moreover, RPMs along the NE reduce the dimensionality of the homolog search from three to two dimensions, thereby further increasing contact probabilities.

After initial contact formation, a subsequent completion of homolog pairing is usually driven by meiotic recombination and formation of a synaptonemal complex (SC) [5]. In many organisms (e.g., budding yeast, mammals and plants), initial recombination processes mediate spatial coalignment specifically of homologous chromosomes. Thereafter, installation of the SC, a highly regular and robust structure, links homolog axes more closely all along their lengths. The completion of meiotic recombination in the context of the SC generates a limited number of crossovers (COs) per bivalent. After CO formation, the SC disassembles and chromosomes are further compacted. Homologs separate along their lengths, except at the sites of COs that become evident as chiasmata. COs in combination with distal sister chromatid cohesion keep the homologous chromosomes stably linked as bivalents until anaphase I, when homologs are separated onto opposite spindle poles.

Despite the outstanding potency of meiotic recombination for specific recognition of homologous DNA, there is a surprising number of organisms, in which homolog pairing does not depend on meiotic recombination. In *C. elegans* and in females of *D. melanogaster*, for example, proper pairing and synapsis proceed also in the absence of meiotic DNA double strand breaks. Moreover, in males of *D. melanogaster*, meiosis includes neither meiotic recombination [6] nor SC formation. Achiasmate meiosis has evolved independently at least 25 times in diverse other lineages [7,8] and it is the rule in higher dipteran males. Y chromosome degeneration resulting in a loss of homology between sex chromosomes might drive evolution of recombination-independent mechanisms for sex chromosome conjunction during meiosis, generating a pre-adaptation for a complete loss of meiotic recombination.

How does meiosis in *D. melanogaster* males succeed independent of meiotic recombination and SC? Asymmetric divisions of germline stem cells, residing in a niche at the closed apical tips of the two epithelial testis tubes present in each male, generate a differentiating daughter cell. This gonialblast, enveloped by two post-mitotic somatic cyst cells, continues with progression through four mitotic division cycles with incomplete cytokinesis. The resulting cyst of 16 interconnected spermatocytes grows during progression through six stages, S1-S6 [9]. After the meiotic divisions, cysts with 64 haploid spermatids complete spermiogenic differentiation, which culminates with the individualization and release of mature sperm into the seminal vesicle at the distal end of the testis tubes.

The process of homolog pairing in *D. melanogaster* spermatocytes has been analyzed initially by live imaging of chromosomal insertions of lacO repeat arrays visualized by GFP-lacI-nls [10] and later also by fluorescence in situ hybridization (FISH) [11,12]. Homolog pairing appears to proceed rapidly after completion of the meiotic S phase in S1 spermatocytes.

It is readily conceivable that homolog pairing in early spermatocytes is driven by the same mechanisms that are also responsible for the pervasive pairing of homologous chromosomes in somatic cells characteristically observed in Dipterans [13,14]. Pairing of homologs in somatic cells is initiated in embryogenesis [15,16]. FISH analyses with embryos have indicated that the pairing of homologous chromosomes is largely abolished during progression through mitosis, followed by rapid restoration within the first 20 minutes of interphase [16]. Overall the most recent studies [17–19] support a model for homolog pairing based on "buttons", i.e. high affinity regions interspersed along the chromosomes where pairing is initiated followed perhaps by zipper-like spreading. Tight-pairing button regions are enriched in architectural and insulator proteins [17,18,20]. Thus, homologs might find each other and pair up because of preferred interactions between allelic and hence identical button-specific combinations of these proteins.

Homolog pairing in cultured *Drosophila* cells has been exploited for genome-wide RNAi screens for factors that either promote or antagonize homolog pairing [21,22]. Beyond many novel candidate regulators, condensin II subunits were identified, which had already been established as potent pairing regulators by compelling analyses with mutants flies [23–25].

Condensin II is a member of the SMC complexes that control genome organization in pro- and eukaryotes [26–29]. SMC proteins form the core of these protein complexes. Three distinct SMC heterodimers were presumably present already in the primordial eukaryote. A SMC1/3 heterodimer forms the basis of the cohesin complexes, SMC2/4 the condensin complexes and SMC5/6 the third complex type. SMC heterodimers combine with a kleisin subunit. The N- and C-terminal domains of the kleisin bind to the head domains of the first and second SMC subunit, respectively. Condensin I and II contain the same SMC2/4 heterodimer but distinct kleisins and additional subunits of the Hawk protein family. In case of condensin I, the Hawks Cap-D and Cap-G are recruited by the γ-kleisin Cap-H, while the Hawks Cap-D3 and Cap-G2 are bound by the β-kleisin Cap-H2 in condensin II.

Condensins were originally identified as abundant chromosomal proteins required for chromosome condensation at the start of M phase during mitotic and meiotic divisions [30–32]. Condensin I is cytoplasmic during interphase but associates with chromosomes after nuclear envelope breakdown (NEBD). In contrast, condensin II localizes to the nucleus already during interphase and can act on chromosomes throughout the cell cycle [33,34]. Condensins appear to promote chromosome compaction by DNA loop extrusion, an activity demonstrated in vitro with both purified condensin and cohesin [35–37]. Re-organization of chromosomal DNA by loop extrusion provides an elegant solution for the topological problems that necessarily arise from the enormous length of chromosomal DNA [29,38,39]. Loop extrusion by SMCs acts in cis exclusively on a given chromatid. In cooperation with topoisomerase II activity, it can generate topologically isolated chromatin domains and even entire individualized sister chromatids free of catenation [40]. Experimental support for the importance of condensin-mediated loop extrusion for compaction and resolution of sister chromatids at the start of M phases is rapidly increasing [41–45]. During mitotic chromosome condensation, the two condensin complexes appear to have somewhat distinct roles. Condensin II provides rigidity by establishing a compact longitudinal axis, whereas condensin I mediates predominantly lateral condensation [32,46,47].

Initial insights into interphase functions of condensin II came from phenotypic characterization of mutations in *Cap-H2* and *Cap-D3* in *D. melanogaster* [23–25]. Analyses with cultured *Drosophila* cells provided further confirmation and insights [17,22,48,49]. Based on genetic interactions, Cap-H2 and Cap-D3 appear to function in a complex with SMC2/4. Observations made in polyploid cells (nurse cells in adult ovaries and cells in larval salivary glands) and cultured cells agree that condensin II activity controls the extent of homolog pairing. Reduced Cap-H2 levels result in more extensive pairing, while increased levels disrupt pairing. Disruption of pairing by condensin II appears to result from axial compaction of interphase chromosomes [25,49].

At the organismal level, *D. melanogaster* condensin II appears to be most important in spermatocytes for normal male meiosis. Severe mutations in *Cap-H2* and *Cap-D3* allow development of morphologically normal adults, which are fertile when female, but sterile when male [23]. Given its anti-pairing activity, why should condensin II be most critical in spermatocytes, where homologs need to pair up for regular segregation in M I? We hypothesize that this apparent paradox arises from the necessity of stabilization of homolog pairing after its initial establishment in spermatocytes. In somatic cells, homolog pairing is effectively disrupted by chromosome condensation and spindle forces at the onset of mitosis. In meiosis, however, premature disruption of bivalents must be avoided. COs stabilize bivalents until anaphase I in females. An alternative homolog conjunction (AHC) system is used in spermatocytes [50–53]. Several genes (*teflon*, *mnm*, *snm* and *uno*) are known to be required specifically for AHC. MNM, SNM and UNO appear to function in a physical linkage that prevents premature separation of bivalents. The molecular details of how these proteins assemble on paired chromosomes and assure linkage are not yet understood. Similarly, it remains to be explained why these linkages are not also established between non-homologous chromosomes. We presume that condensin II and the remarkable process of chromosome territory formation are critical in this regard. Chromosome territory formation occurs during the S2b stage. DNA staining after territory formation reveals three separate regions within the spermatocyte nucleus. One of these regions contains the bivalent formed by the large autosome chromosome (chr) 2, the bivalent of the other large autosome chr3 is in a second territory, and the third territory hosts the sex chromosome bivalent. The *D. melanogaster* karyotype includes one additional autosome, chr4, which is a small dot chromosome, and its bivalent is frequently associated with the chrXY territory. Evidently, establishment of AHC after chromosome territory formation

would minimize the danger of inappropriate linkage between non-homologous chromosomes. AHC establishment while chromosomes are still intermingled might link homologous as well as non-homologous chromosomes.

The forces that separate bivalents apart during chromosome territory formation in spermatocytes are not known, but condensin II proteins are important for this process [23]. *Cap-D3* and *Cap-H2* mutant spermatocytes do not display chromosome territories after stage S2b. For further clarification of the mechanisms that separate bivalents apart during chromosome territory formation, we have applied time-lapse imaging. Using established fluorescent centromere and telomere markers, as well as novel probes for pericentromeric heterochromatin, we have characterized the spatial and temporal dynamics of the disruption of non-homologous associations between these chromosomal regions during territory formation. The effects of inhibitors of F actin and microtubules were analyzed. Moreover, newly generated *Cap-H2* alleles were studied as well, including the contribution of AHC to chromosome missegregation observed in condensin II mutants during M I. Our findings indicate that chromosome territory formation does not involve mechanisms analogous to those that drive RPMs in other species. While our work provides further support for the importance of condensin II and control of Cap-H2 levels, we report findings that suggest the involvement of yet unidentified additional forces in chromosome territory formation.

## Results

### Spatial dynamics of centromeres and telomeres during chromosome territory formation

To study the dynamics of chromosome territory formation, we performed time-lapse imaging of spermatocytes expressing His2Av-mRFP and Cenp-A/Cid-EGFP. His2Av-mRFP resulted in weak diffuse signals throughout the nucleus and in stronger signals in regions with chromatin (Fig 1A). Based on the diffuse signals, the nuclear diameter ($d_n$) was determined, which correlates with the spermatocyte developmental stage [9,10]. In spermatocytes with $d_n$ around 9 μm, i.e. during late S2 stage, our time-lapse imaging revealed the conversion of the His2Av-mRFP chromatin signals from a state without to a state with spatially separated chromosome territories (Fig 1A), as expected [9]. The impossibility to delineate individual bivalents before territory formation precluded a precise scoring of the start of the relatively gradual process. However, to a first approximation, the spatial re-organization of chromatin appeared to take about three hours in the fastest cases and up to about six hours, during which nuclear diameters grew minimally (1–4%).

Time-lapse imaging of the centromeric Cenp-A/Cid-EGFP dots indicated that chromosome territory formation was accompanied by centromere de-clustering (Fig 1A and 1B and S1 Movie), as implied previously by live imaging at single time points [10] and by microscopy with fixed samples [54]. The overwhelming majority of S1/2 spermatocytes, which did not yet have well separated territories according to the His2Av-mRFP signals, displayed 2–3 Cid-EGFP dots (97%, n = 184). Spermatocytes with either one or more than three dots at these early stages were rare (2 and 1%, respectively). A series of dot splitting events was observed during chromosome territory formation and thereafter, raising dot number up to six or seven (Fig 1A and 1B). More rarely, dot fusions were observed as well, which reduced the total dot number but only transiently. The temporal dynamics of centromere de-clustering varied considerably (Figs 1C and S1). The four-dot stage had the lowest variability and the shortest mean duration compared to the three- and five-dot stages (Fig 1C). Based on the spermatocytes progressing from the three- to the five-dot stage during the 12 hour imaging period, the mean duration of the four-dot stage was 72 min (± 50 s.d., n = 15) (Fig 1C). This is a slight

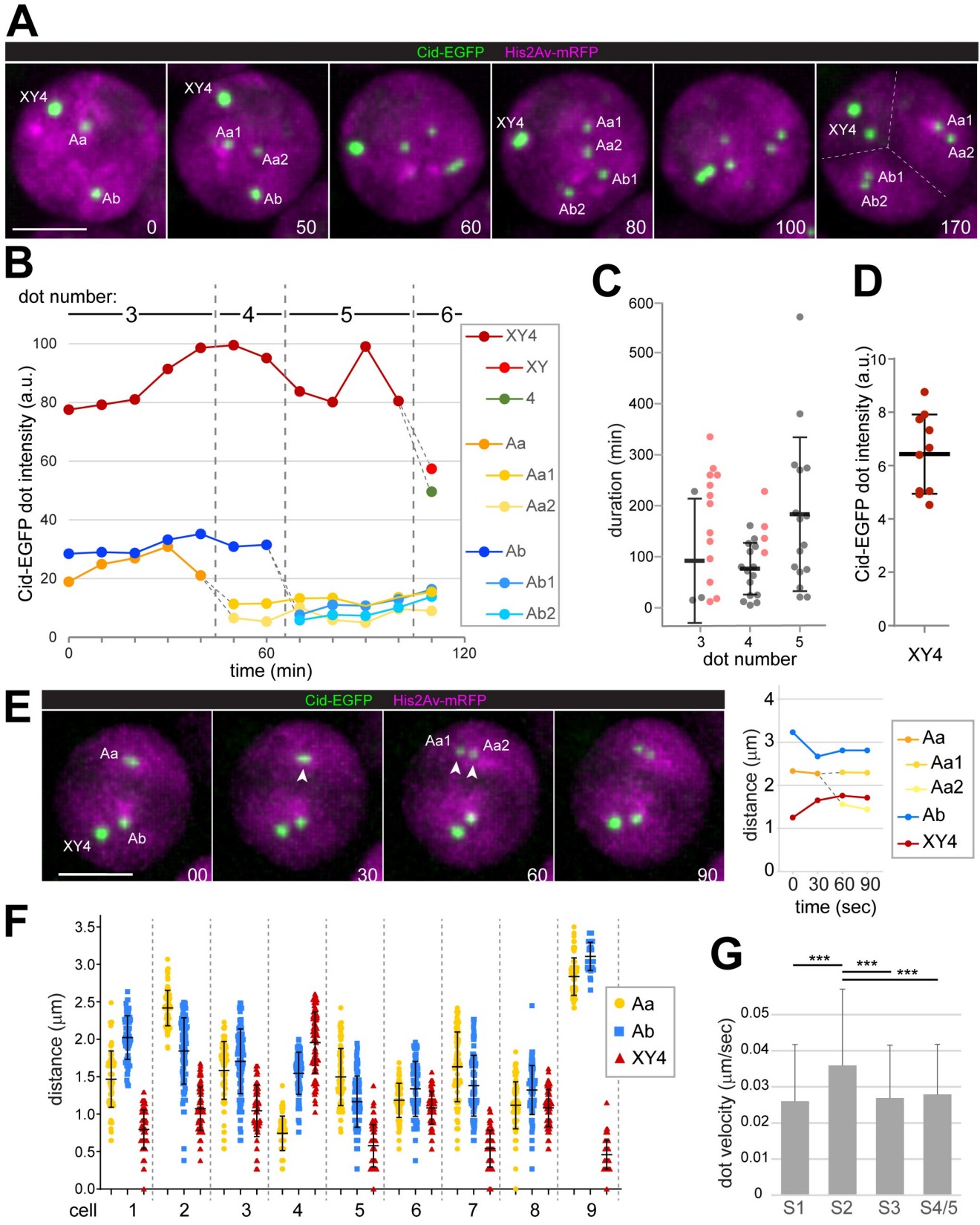

**Fig 1. Disruption of centromere clusters during chromosome territory formation.** Spermatocytes expressing His2Av-mRFP and Cenp-A/Cid-EGFP were analyzed by time-lapse imaging. (**A**) Labeling of Cid-EGFP dots indicates their association with distinct chromosomes. "Aa" and "Ab" designate centromere clusters associated with the large autosomes (chr2 and chr3) with "1" and "2" denoting the two homologs. "X", "Y" and "4" designate the clustered centromeres of the additional chromosomes (chrX, chrY and chr4). Complete separation of chromosome territories (dashed lines) is evident at the last time point. Scale bar = 5 μm. (**B**) Signal intensity of the Cid-EGFP dots displayed by the spermatocyte shown in (A). Dashed grey lines indicate dot-splitting events. (**C**) Duration of the stages with three, four and five Cid-EGFP dots per nucleus, respectively. Grey values represent actual stage durations determined for spermatocytes that progressed through both dot-splitting events delimiting a particular stage. Mean ± s.d. of these values is displayed as well. Values plotted in color are from cells, in which only one of the two delimiting dot-splitting events was observed during the imaging period, thus indicating minimal durations. (**D**) Relative Cid-EGFP signal intensity of the XY4 dot at the four-dot stage compared to the Aa1 dot (formed by the two tightly associated, unresolved sister centromeres of a large autosome). Mean ± s.d. is displayed as well (n = 10). (**E**) Cid-EGFP dots are far from the nuclear periphery during dot-splitting events. A representative event (Aa into Aa1 and Aa2) with a graph indicating the shortest distances between dots and nuclear periphery is shown. t = 0: start of stretching of the separating Cid-EGFP dot. (**F**) Position of Cid-EGFP dots relative to the nuclear periphery were tracked (after time-lapse imaging at five-second intervals over 7.5 min). Nine spermatocytes with three dots (Aa, Ab and XY4) were analyzed. Separation distance of a given Cid-EGFP dot at each time point (n = 89) is plotted, as well as mean ± s.d. (**G**) Cid-EGFP dots move faster during the S2 stage when chromosome territories form. Cid-EGFP dots were tracked (after time-lapse imaging at five-second intervals). Bars represent mean dot velocity (± s.d.) obtained by averaging over all dots present in a cell (i.e., 3–5 dots depending on stage) over all 99 time intervals and over all cells analyzed for a given stage (3–5 spermatocytes, see S3 Fig). *** indicates p < .001 (t test). Scale bars = 5 μm.

underestimate, because there were a few among the imaged cells with a more extended four-dot stage. The actual duration of the four-dot stage could not be determined in these exceptional cells, because only one of the two dot-splitting events delimiting the four-dot stage was observed within the imaging period (12 hours). In case of the three-dot stage, only three among the analyzed cells displayed both delimiting dot-splitting events during the imaging period. However, cells progressing through only one of the dot-splitting events provided additional confirmation that the three-dot stage lasts longer than the four-dot stage.

Centromere de-clustering was paralleled by changes in signal intensities of the centromeric Cid-EGFP dots. Intensity quantification in combination with the tracking of centromeric signals over time indicated that the four-dot stage rarely represented a state where each bivalent is associated with a single centromere cluster. During entry into the first meiotic division, when each of the four bivalents can be identified, centromeres display highly comparable Cid-EGFP intensities except for the chrY centromere, which exhibits a twofold higher intensity [54,55]. Accordingly, if each dot at the four-dot stage represents a cluster of all the centromeres of a bivalent, the dot on the sex chromosome bivalent is expected to have a 1.5 fold greater intensity than the remaining three autosomal dots of equal intensity. However, the intensities at the four-dot stage did not conform to this prediction. Compared to the weakest centromere dot, the strongest was on average 6.4 fold stronger rather than just 1.5 fold more intense (Fig 1D). Moreover, dot tracking over time revealed that the dots present at the four-dot stage were not partitioned each into a distinct chromosome territory. In the majority of the analyzed spermatocytes (85%, n = 20), the two weakest dots of the four-dot stage were partitioned together into one of the two large autosomal chromosome territories (Fig 1A). These large autosomal territories were characterized by more uniform His2Av-mRFP signals compared to the territories of chr4 and chrXY, which were usually in close spatial association [54,55].

Back-tracking centromeric dot signals from stages after territory formation to earlier stages revealed a most frequent centromere de-clustering program. When a single centromere cluster was present at the earliest stage, a first dissociation event liberated a smaller centromere cluster (Aa), containing all centromeres of one of the large autosomal bivalents (i.e., a cluster with the two homologous pairs of sister centromeres). A subsequent event dissociated an analogous cluster with all centromeres of the other large autosome (Ab) from the most intense centromere cluster. This most intense cluster thereafter still contained all centromeres of chrX, Y and 4. Accordingly, centromere clusters Aa, Ab and XY4 were present at the three-dot stage (Fig 1A). Importantly, the subsequent four-dot stage did not result from XY4 cluster splitting into a chrXY and a chr4 cluster. Rather, the Aa centromere cluster was separated into two dots (Aa1

and Aa2), each containing an unresolvable sister centromere pair (Fig 1A). Typically, chromosome territory formation paralleled the transition from the three—to the four-dot stage. The subsequent splitting of the Ab cluster into two dots (Ab1 and Ab2) resulted in the five-dot stage (Fig 1A). The six- and seven-dot stages arose by release of dots from the most intense XY4 cluster. At the seven-dot stage, each of the large autosome territories Aa and Ab contained two dots, and the XY4 territory comprised three additional dots: the paired sister centromeres of chrX (dot 1) and of chrY (dot 2), and a dot with all the chr4 centromeres (dot 3). A final de-clustering step dissociated the single chr4 dot into two, each containing a pair of sister centromeres. This final transition to the eight-dot stage occurred usually after a long delay just around the time of NEBD I at the end of the S6 stage [55]. A permanent separation of sister centromeres before M I was never observed, but occasional transient breathing of sister centromeres in large autosomal territories could be detected, as described [55].

The de-clustering program described above was observed in 65% of the spermatocytes analyzed by dot tracking (n = 20). The remaining 35% displayed deviations from the canonical program. In 20%, the variation concerned progression from the four to the five-dot stage. In these cases, the five-dot stage was reached by the release of a dot from the most intense XY4 dot before Ab splitting. The newly released dot contained all centromeres of either chrY or chr4, because it was about twofold more intense than Aa1 or Aa2. The final 15% of the tracked spermatocytes varied in the progression from the three to the four-dot stage. In these cells, the release of a dot from the most intense XY4 dot occurred even earlier, already at the three-dot stage before Aa splitting. Again, the dot released early represented all centromeres of either chrY or chr4 based on its intensity. Of relevance for Cid-EGFP dot intensity quantification, the Aa dot splitting into Aa1 and Aa2, as well as the analogous Ab dot splitting and the occasional transient sister centromere breathing events were all accompanied by an approximate 1:1 partitioning of the precursor dot intensity (Fig 1B), indicating that Cid-EGFP quantification permits a reliable detection of twofold differences, in particular when positions along the z axis remained comparable.

Overall, these results disprove the presence of an obligate four-dot stage, where each bivalent displays a single centromere cluster, as previously claimed [10]. Such a centromere pattern is present in at most 15% of the spermatocytes.

The absence of an obligate four-dot stage with a single centromere cluster per bivalent does not necessarily rule out an involvement of centromeres in territory formation. Cytoskeletal forces might still act via centromeres to increase chromosome mobility in scenarios more complicated than a straightforward separation of bivalents via unique handles. However, if territory formation were to involve centromere movements by cytoskeletal forces acting via LINC complexes (or comparable linkages), centromeres would be expected to move at close distance along the NE. Therefore, we analyzed the intranuclear positions of the Cid-EGFP dots. First, we focused on dot de-clustering events (two-to-three and three-to-four dot transitions). In six out of eight analyzed events, dot splitting occurred far from the nuclear periphery. The splitting dot was between 0.4 and 2.3 μm away from an isosurface delineating the nuclear His2Av-mRFP signal (Fig 1E). Cid-EGFP dots present in addition to the splitting dot were also localized far from the nuclear periphery (Fig 1E).

For further analysis of Cid-EGFP dot positions, spermatocytes were imaged at 5 sec intervals for error-free dot tracking at high resolution. The spatial separation of Cid-EGFP dots from an isosurface delineating the nuclear His2Av-mRFP signal was determined in spermatocytes at the S2 stage when chromosome territories form. As described above, three Cid-EGFP dots were usually present at this stage. Cid-EGFP dots at positions close to the nuclear periphery (d < 0.4 μm) were almost never observed (Fig 1F), even in case of the XY4 dot, which tended to be closer to the nuclear periphery than the Aa and Ab dots (Fig 1F). Finally, we

labeled Cid-EGFP whole mount testis preparations with anti-Lamin and a DNA stain. Centromere positions were found to be at least 0.5 μm and up to 3.5 μm away from the nuclear lamina during the stages S1 to S5 (S2 Fig). Moreover, at the stages before complete separation of the territories, DNA signals were separated from the nuclear lamina by an unstained gap (S2 Fig). In conclusion, given the considerable distance separating centromeres from the nuclear envelope, a direct mechanical coupling between cytoskeletal forces and centromeres for the purpose of territory formation appears improbable.

The Cid-EGFP dot tracking was also used for the characterization of centromere movements. After correcting lateral nuclear drift (using an isosurface around the His2Av-mRFP signal), centromere dot velocities were determined. Similar analyses with *Drosophila* ovaries have revealed rapid nuclear rotations in gonial cells during the stage where the cysts consist of eight cells [3,56]. In testes, the average speed of centromeres was around 0.03 μm/sec during interphase in gonial eight-cell cysts, as well as in S1-S5 spermatocytes (Figs 1G and S3). This is tenfold slower than during the nuclear rotations in ovarian eight-cell cysts [3,56], and also considerably slower than the RPMs during *C. elegans* meiosis [57]. Interestingly, however, in *Drosophila* spermatocytes at the S2 stage, transient phases of more rapid movements in particular of the most intense XY4 centromere cluster tended to be more frequent than in the preceding and following stages (S3 Fig). Averaging across all Cid-EGFP dots, all time points and all cells confirmed that centromere movements were significantly faster during the S2 stage compared to the stages before and thereafter (S1, S3, S4/5) (Fig 1G). Therefore, territory formation, which occurs during the S2 stage, is accompanied by phases with increased chromosome mobility. Similar phases with comparatively fast movements of at least some centromeres were observed in early round spermatids (S3 Fig), in which up to four Cid-EGFP dots were detectable. The increased centromere mobility in spermatids coincides with the known, striking relocalization of LINC and nuclear pore complexes from a spherical to a hemispherical distribution within the NE of these cells [58,59].

In mammals and yeast, RPMs during meiosis are led by telomeres linked via LINC complexes to cytoskeletal forces [3,4]. In *Drosophila*, electron microscopy has failed to reveal a classical bouquet stage with telomeres attached or close to the NE in pachytene oocytes [60]. To characterize telomere behavior in spermatocytes, we exploited an EGFP knockin at *caravaggio* (*cav*) [61]. This gene codes for the telomere protein HOAP. If each telomere were resolved as a dot, the number of EGFP-HOAP dots per nucleus would be 16 in G1 and 32 in G2. However, only four to six EGFP-HOAP dots were detected in early embryos and larval brain cells, as a result of telomere clustering [62]. Based on limited data from fixed spermatocytes, partial telomere clustering occurs in these cells also [61,63]. For a detailed analysis, we applied time-lapse imaging. Analysis of M I confirmed that EGFP-HOAP is a faithful telomere marker in spermatocytes (Fig 2A). EGFP-HOAP dots were localized at the trailing ends of the large chromosomes during anaphase I. Counts of the EGFP-HOAP dots provided further support that they represent telomeres. At the late S6 stage, after partial condensation of bivalents into well-isolated territories just before NEBD I, 25 EGFP-HOAP dots were present in the cell shown for illustration (Fig 2A). Eight dots were present in one of the large autosome territories (Fig 2A), corresponding to the expected number in case of separated sister telomeres. Only seven EGFP-HOAP dots were resolved in the other large autosome territory (Fig 2A), because two were presumably too close. Finally, ten EGFP-HOAP dots were associated with the XY4 territory, and two of those were clearly brighter than the rest (Fig 2A). Each of these two bright dots contained the four clustered telomeres of a chr4 homolog, as indicated by their subsequent segregation during M I. Numbers and distributions of EGFP-HOAP dots were comparable in the analyzed M I spermatocytes (n = 7 from five distinct cysts). In conclusion, close to all

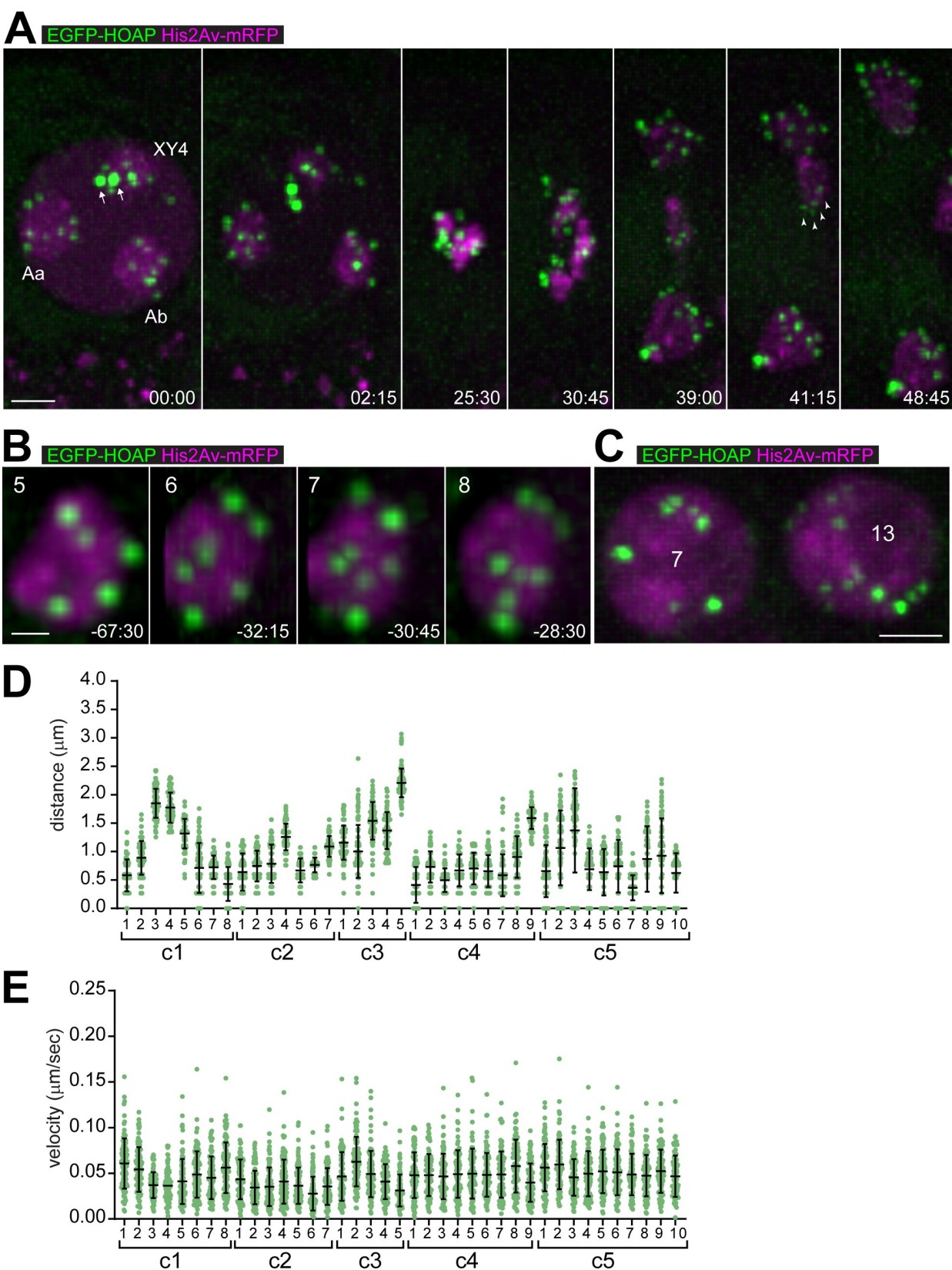

**Fig 2. Telomeres display limited movements far from the nuclear periphery during territory formation.** Spermatocytes expressing His2Av-mRFP and the telomere protein EGFP-HOAP were analyzed by time-lapse imaging. (**A**) Progression through M I. Time (min:sec) is given relative to the onset of NEBD I. Chromosome territories labeled as in Fig 1A. The two most intense EGFP-HOAP dots (arrows) represent telomere clusters of the two homologs of the small dot-like chr4. During anaphase, EGFP-HOAP dots were clearly at the trailing ends of long chromosome arms (see arrowheads for example). (**B**) Telomere de-clustering at the end of the S6 stage. A large autosome territory is shown with the number of EGFP-HOAP dots indicated (top). Time (min:sec) is given relative to the onset of NEBD I. (**C**) During the stage of chromosome territory formation, telomeres are partially clustered. EGFP-HOAP dots vary in numbers and intensity, as illustrated with two neighboring spermatocyte nuclei from an S2 cyst with the number of EGFP-HOAP dots per nucleus indicated. (**D,E**) Intranuclear positions and velocities of telomeres during territory formation. After time-lapse imaging (at five-second intervals), all EGFP-HOAP dots sufficiently strong for reliable tracking over time were analyzed in five distinct cells (c1-c5) during the S2 stage. All distances (**D**) and all velocities (**E**) observed for a given EGFP-HOAP dot during an 8.5 min period were plotted (n = 102), as well as the mean ± s.d. Scale bars = 3 μm (A), 1 μm (B) and 3 μm (C).

telomeres can be resolved individually in late spermatocytes except for those of the small chr4. Moreover, the amount of EGFP-HOAP per telomere does not vary extensively.

Time-lapse imaging at earlier stages revealed fewer EGFP-HOAP dots of more variable intensities, indicating partial telomere clustering. Indeed, de-clustering of EGFP-HOAP dots could be observed readily during the S6 stage (Fig 2B). In early spermatocytes during the stages of chromosome territory formation, the number of telomere clusters was variable, as well as the intensities of the associated EGFP-HOAP signals. Even neighboring cells within the same cyst were observed to differ with regard to the extent of telomere clustering (Fig 2C). The EGFP-HOAP dots in S2 spermatocytes were tracked. As for the Cid-EGFP signals, the distance between telomere dots and an isosurface delineating the nuclear His2AvD-mRFP signal was determined, as well as their speed. Only the stronger EGFP-HOAP dots that could be tracked reliably were considered. Our results revealed that telomere clusters are neither consistently close to the NE (Fig 2D) nor moving fast (Fig 2E) during the stages of chromosome territory formation. Therefore, telomeres are unlikely used for the mobilization of chromosomes by mechanisms comparable to those acting during the bouquet stage in yeast, mammals, or *C. elegans* [3,4].

## Disruption of centromere clusters and chromocenters during chromosome territory formation in spermatocytes requires Cap-D3 and Cap-H2

While centromeres and telomeres do not appear to have a prominent role during chromosome territory formation in *Drosophila* spermatocytes, condensin II (Fig 3A) was suggested to be crucial for this process [23]. Previous analyses of *Cap-D3* and *Cap-H2* function were completed with mutant alleles that might not necessarily cause a complete loss of function. While the presence of several genes within the first *Cap-D3* intron complicates the isolation of null mutations, this appeared feasible in case of *Cap-H2*. Applying CRISPR/Cas9 in combination with a pair of single guide RNAs for the generation of intragenic deletions of central *Cap-H2* regions led to three distinct alleles (Fig 3B). *Cap-H2*[cc1] and *Cap-H2*[cc2] are intragenic, in-frame deletions of 918 and 1488 bp, respectively. *Cap-H2*[cc3] carries a 5 bp, frame-shifting deletion in the gRNA1 target region, as well as a 918 bp deletion further downstream. These newly induced alleles are expected to affect all four annotated Cap-H2 isoforms (D-G) (Fig 3B). In wild type, the isoforms E, F and G are identical except for a short internal stretch of 13 amino acids (aa) that is either present (E), absent (F), or shortened to 12 aa (G), as a result of differential splicing of the fourth intron. The D isoform is most unusual. It is a variant F isoform without the N terminal-most region and with a distinct C terminal region, resulting from the use of a downstream transcriptional start site and differential splicing. The newly isolated *Cap-H2*[cc3] mutation might still permit expression of N-terminal Cap-H2 fragments (aa 1–186 of E-G; aa 1–8 of D) followed by a few extra amino acids after the frame shift until the premature stop. These truncated *Cap-H2*[cc3] products lack the regions predicted to bind Cap-D3, Cap-G2

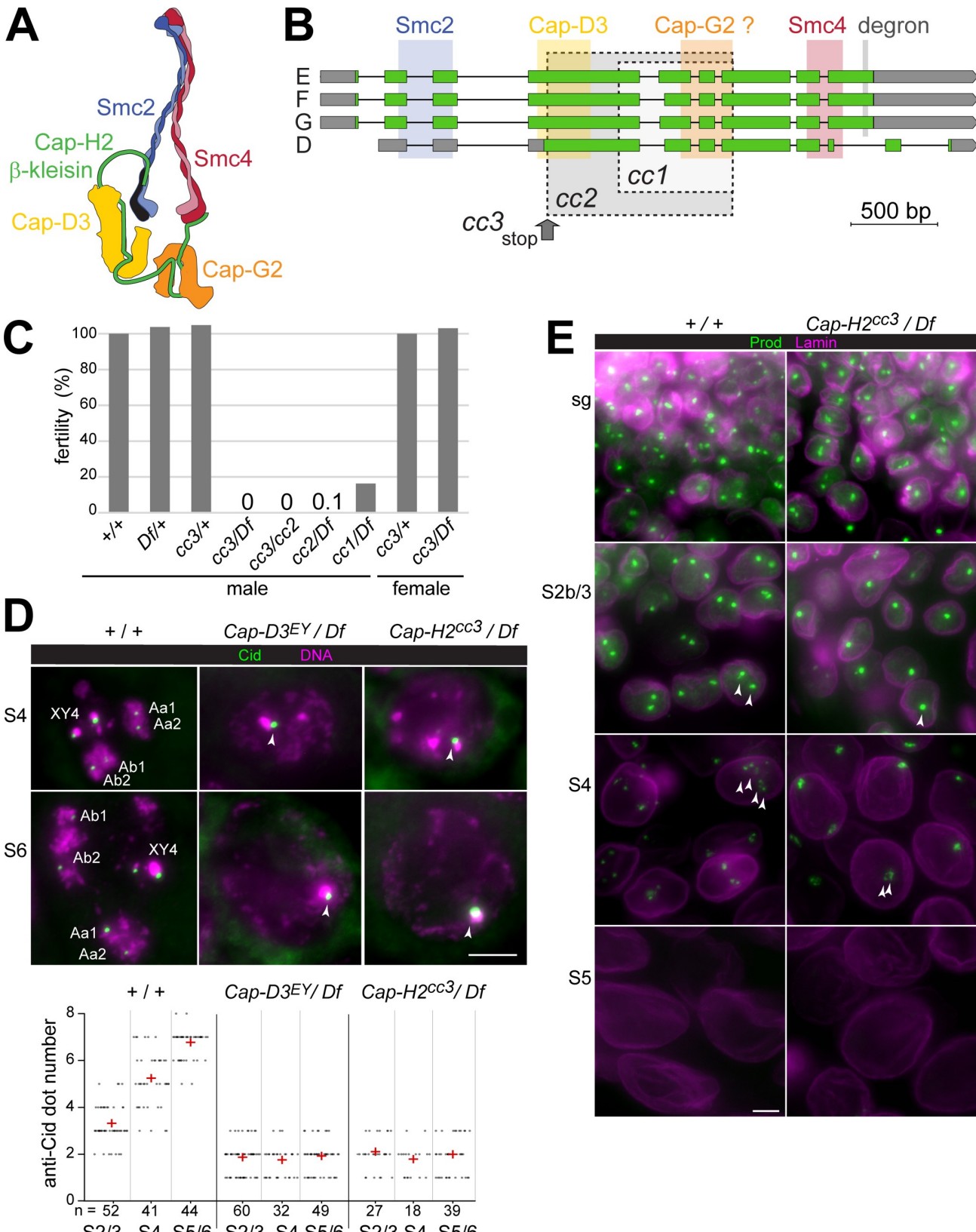

**Fig 3. Intragenic *Cap-H2* deletions abolish disruption of centromere clusters and chromocenters in spermatocytes.** (**A**) Condensin II complex organization. (**B**) *Drosophila Cap-H2* codes for four isoforms annotated as E, F, G and D. Predicted binding sites for condensin II subunits are indicated, as well as a degron [48]. *Drosophila* Cap-G2 existence is uncertain (see discussion). Three intragenic deletions, *cc1 –cc3*, were generated by CRISPR/Cas9. Regions deleted in case of the in-frame deletions *cc1* and *cc2* are indicated (grey shading with dashed borders). Two deletions are present in *cc3*, a small 5 bp deletion resulting in a premature stop (grey arrow) and a downstream in-frame deletion identical as in *cc1*. (**C**) Fertility of adult flies with the indicated genotypes. The deficiency *Df(3R)Exel6159* (*Df*) eliminates *Cap-H2*. Bars report male fertility relative to +/+ males (100%), and female fertility relative to *cc3*/+ (100%). (**D**) Failure of centromere cluster disruption. Squash preparations of testes from control (+ / +) and the indicated *Cap-D3* and *Cap-H2* mutants were labeled with anti-Cid and a DNA stain. High magnification views of spermatocyte nuclei at the indicated stages are displayed. The dot plot presents the number of anti-Cid dots per nucleus, as well as the mean dot number (red cross) at the indicated stages. (**E**) Failure of chromocenter disruption. Whole mount preparations of testes from the indicated genotypes were labeled with anti-Prod and anti-Lamin Dm0. High magnification views from regions with spermatogonial cells (sg) and spermatocytes at the indicated stages are presented. Scale bars = 5 µm.

and SMC4/Gluon (Fig 3B) [64–66], but the N-terminal SMC2-binding region is still present. Thus, the putative truncated E-G isoforms encoded by *Cap-H2*[cc3] might exert a dominant-negative effect in principle, although mitigated by non-sense mediated RNA decay presumably. The protein products encoded by *Cap-H2*[cc1] and *Cap-H2*[cc2] include both the N- and the C-terminal domain for binding to SMC2 and SMC4, respectively (Fig 3B). However, these internally truncated β-kleisins are predicted to be defective in Hawk recruitment. In case of *Cap-H2*[cc2], the regions predicted to bind Cap-D3 and Cap-G2 are both absent, while only the former is missing in *Cap-H2*[cc1] (Fig 3B).

Zygotes with either one of the three newly isolated *Cap-H2* alleles in trans over a deficiency deleting *Cap-H2* developed to the adult stage with Mendelian frequencies. Morphologically, the hemizygous adults were apparently normal and females were fully fertile (Fig 3C). In contrast, male fertility was severely reduced in case of *Cap-H2*[cc1], almost completely eliminated by *Cap-H2*[cc2] (one escaping progeny fly), and totally abolished by *Cap-H2*[cc3] (Fig 3C). *Cap-H2*[cc1], therefore, and perhaps even *Cap-H2*[cc2] have residual function. Moreover, we conclude that *Cap-H2* function is essential exclusively for male fertility, and it appears to contribute in a non-essential manner to cellular processes like chromosome condensation during M phase [67], interphase chromosomal organization and gene expression [17,22,24,49]. In contrast to *Drosophila*, the mouse ortholog *Ncaph2* is required for embryonic development [68].

For an initial analysis of the effects of the newly isolated *Cap-H2* alleles on chromosome territory formation, we analyzed squash preparations of testes after DNA staining. The characteristic compartmentalization of nuclear DNA staining that reveals chromosome territories in control spermatocytes at stage S2b and later, was never observed in *Cap-H2*[cc3]/ *Df(3R) Exel6159* mutants (S4 Fig), as previously reported for other strong *Cap-H2* and *Cap-D3* alleles [23]. The phenotypes observed in *Cap-H2*[cc3]/ *Df(3R)Exel6159* and *Cap-D3*[EY00456]/ *Df(2L) Exel7023* were indistinguishable (S4 Fig). As *Cap-H2*[cc3], *Cap-D3*[EY00456] is very likely an amorphic allele [23]. The other alleles, *Cap-H2*[cc1] and *Cap-H2*[cc2], also resulted in severe defects in chromosome territory formation (S4 Fig).

To clarify whether *Cap-D3* and *Cap-H2* are required for the disruption of centromere clusters during chromosome territory formation and subsequent spermatocyte maturation, we applied anti-Cenp-A/Cid labeling (Fig 3D). In control spermatocytes at the S2/3 stage, when chromosome territories are formed, an average of 3.3 (± 0.8 s.d.) dots were detected by anti-Cid (Fig 3D). At stages S4 and S5/6, the number of dots increased to 5.2 (± 1.2 s.d.) and 6.7 (± 0.6 s.d.), respectively (Fig 3D). However, in spermatocytes of *Cap-D3* and *Cap-H2* mutant males (*Cap-D3*[EY00456]/ *Df(2L)Exel7023* and *Cap-H2*[cc3]/ *Df(3R)Exel6159*) centromere de-clustering was not observed. The mean number of anti-Cid dots per nucleus remained essentially constant at around two (Fig 3D). This result strongly argues against the notion that chromosome territories might still be formed in condensin II protein mutants, remaining cryptic however when analyzed by standard DNA staining because of extensive territory overlap resulting

from a chromatin condensation failure. Evidently, the elimination of non-homologous chromosome associations is largely blocked in the absence of condensin II proteins.

Recently, the heterochromatin proteins D1 and Prod were shown to be crucial for chromocenter formation in *Drosophila* cells [69,70]. These proteins bind to pericentromeric satellites. D1 binds primarily to the {AATAT}$_n$ satellite, which is abundant on chrX, Y and 4 but scarce on chr2 and 3. However, chr2 and 3 contain extensive arrays of the {AATAACATAG}$_n$ satellite, which recruits Prod. Dynamic binding of D1 and Prod to their target satellites, in combination with dynamic interactions between these two proteins, was proposed to drive intranuclear clustering of pericentromeric heterochromatin of all chromosomes into a chromocenter [69,70]. Conversely, chromosome territory formation in spermatocytes, which disrupts the non-homologous chromosome associations in the chromocenter, might depend on control of D1 and Prod behavior. A re-distribution of D1 and a disappearance of Prod was reported to accompany chromosome territory formation during normal spermatogenesis [70]. Moreover, experimental prolongation of Prod expression was shown to result in incompletely separated, bridged chromosome territories [70], suggesting that the normal disappearance of Prod during spermatogenesis is crucial for normal territory formation. However, it remains to be explained why D1 is re-distributed and why Prod disappears from spermatocytes around the stages of chromosome territory formation. In principle, these processes might be driven by condensin II.

To evaluate whether condensin II proteins are required for displacing D1 and Prod from their pericentromeric target satellites, we analyzed the behavior of these proteins in *Cap-H2*$^{cc3}$/ *Df(3R)Exel6159* mutants and controls (Figs 3E and S5). In wild-type testes, D1-sfGFP expressed under control of the endogenous *D1* cis-regulatory region was readily detected in the previously reported pattern [70] (S5 Fig). In spermatogonial cells, D1-sfGFP was strongly enriched in 1–3 dots per nucleus. In contrast, in S1/2 spermatocytes, D1-sfGFP signals were reduced and far more diffuse throughout the nucleus. This drastic change in distribution was also observed in *Cap-H2* mutant testis (S5 Fig), indicating that the re-organization of D1 before chromosome territory formation does not depend on Cap-H2.

Prod expression in wild-type testis detected with anti-Prod [71] was also observed essentially in the previously reported pattern [70], although with a clear asynchrony in the re-organization of D1 and Prod during spermatogenesis (S5 Fig). In spermatogonial cells, anti-Prod signals were strongly enriched in 1–2 intranuclear dots. Early spermatocytes during the stages S2b/S3, when chromosome territories are formed, displayed primarily two Prod dots per nucleus (Fig 3E). During S4, anti-Prod signals decreased in intensity and in S5/6 spermatocytes, they were no longer detectable (Fig 3E). In *Cap-H2* mutants, spermatogonial cells displayed anti-Prod signals indistinguishable from wild-type (Fig 3E). However, most S2b/S3 spermatocytes in *Cap-H2* mutants displayed one rather than two anti-Prod dots (Fig 3E). At later stages, Prod disappeared also from *Cap-H2* mutant spermatocytes as in wild-type (Figs 3E and S5). Interestingly, during Prod disappearance, the anti-Prod signals in both control and *Cap-H2* mutants were frequently re-organized in space into a series of finer dots.

In conclusion, these results indicate that condensin II does not promote chromosome territory formation by enforcing the release of D1 and Prod from the chromocenter. Moreover, the disappearance of Prod might be of limited importance for chromosome territory formation. In wild-type, this disappearance occurred primarily after the partitioning of DNA into territories. Anti-Prod dots of maximal intensity were still present during the stages of chromosome territory formation in wild-type S2b/S3 spermatocytes and the disintegration of the anti-Prod dots during disappearance suggested that the separation forces acting on chromatin during territory formation might be greater than a putative Prod-mediated heterochromatin

cohesion. However, the persistence of a single anti-Prod focus in *Cap-H2* mutants during territory formation indicated that successful chromocenter disruption requires Cap-H2.

## Failure of bivalent individualization in *Cap-D3* and *Cap-H2* mutants precludes regular bi-orientation and segregation during M I

Meiotic chromosome missegregation in *Cap-D3* and *Cap-H2* mutant males has been described previously based on segregation analysis of genetic markers and cytology with fixed samples [23]. For further characterization, we performed time-lapse imaging with condensin II mutant spermatocytes expressing Cenp-A/Cid-EGFP and His2Av-mRFP. Analogous analyses of control spermatocytes were reported previously [55]. In control spermatocytes before NEBD I, His2Av-mRFP reveals the three major chromosome territories containing chrXY4, chr2 and chr3, respectively (Fig 4A). Each of the two large autosome territories (Fig 4A, Aa and Ab) almost invariably displays two Cid-EGFP dots. In contrast, the number of Cid-EGFP dots associated with the chrXY4 territory is more variable. In spermatocytes with fewer than four Cid-EGFP dots within the chrXY4 territory, residual centromere clusters are disrupted during NEBD I, when the final rapid chromosome condensation converts the territories into compact chromosome blobs. The His2Av-mRFP marker reveals the large bivalents clearly (chrXY, 2 and 3), but the dot chr4 contains very little His2Av-mRFP (Fig 4A). Rapid centromere-led, saltatory movements of bivalents during prometaphase I precede their eventual bi-polar integration into a compact metaphase plate before bivalents split in anaphase I (Fig 4A).

During the S6 stage, *Cap-H2* mutant spermatocytes (*Cap-H2*cc3/ *Df(3R)Exel6159*) did not display the characteristic chromatin compartmentalization into three major territories. While chromatin was still enriched at the nuclear periphery, it was in a contiguous reticular pattern. Moreover, the large majority displayed only one or two Cid-EGFP dots. These centromere clusters were dissociated into multiple dots (usually around six per nucleus) during the final two hours before NEBD I, in parallel with the initial slow condensation and release of chromatin from the NE that accompanied nuclear rounding (Fig 4B). However, the individualized Cid-EGFP dots remained in close proximity during NEBD I, when chromatin was rapidly compacted into a single large clump (Fig 4B). Remarkably, this final rapid chromosome condensation during NEBD I, proceeded with apparently normal efficiency, presumably driven by condensin I. During prometaphase I, the single chromatin clump was partly broken up, when centromeres along with associated chromatin were pulled away by spindle forces. The chr4 bivalent was most often separated away (Fig 4C), indicating that this small chromosome was less stably detained within the chromatin clump compared to other bivalents. Released bivalents were usually bi-oriented eventually, moving into a metaphase plate that also contained the residual chromosome clump. During anaphase, centromeres were pulled towards the spindle poles (Fig 4B) and chromosomes were separated with some lagging and with variable success. 44% of the spermatocytes (n = 63) displayed chromatin bridges during anaphase (Fig 4G). These chromatin bridges were eventually resolved during telophase in some cells, but also persistence of bridges and micronuclei formation was observed in 22% of the spermatocytes (n = 63) (Fig 4D) or a complete failure of chromatin clump separation into two portions in 11% of the spermatocytes (n = 63) (Fig 4E).

Bi-orientation of centromere pairs during prometaphase I was severely compromised in *Cap-H2* mutants. As an estimate for the upper limit of the bi-orientation success, the fraction of spermatocytes with a 4:4 separation of centromeres during anaphase I was determined. The chromatin clumping in the mutants precluded an assignment of Cid-EGFP dots to specific bivalents, and thus an unknown number of the 4:4 separation events might reflect balanced syntelic rather than regular amphitelic chromosome attachment. In *Cap-H2*cc3/ *Df(3R)*

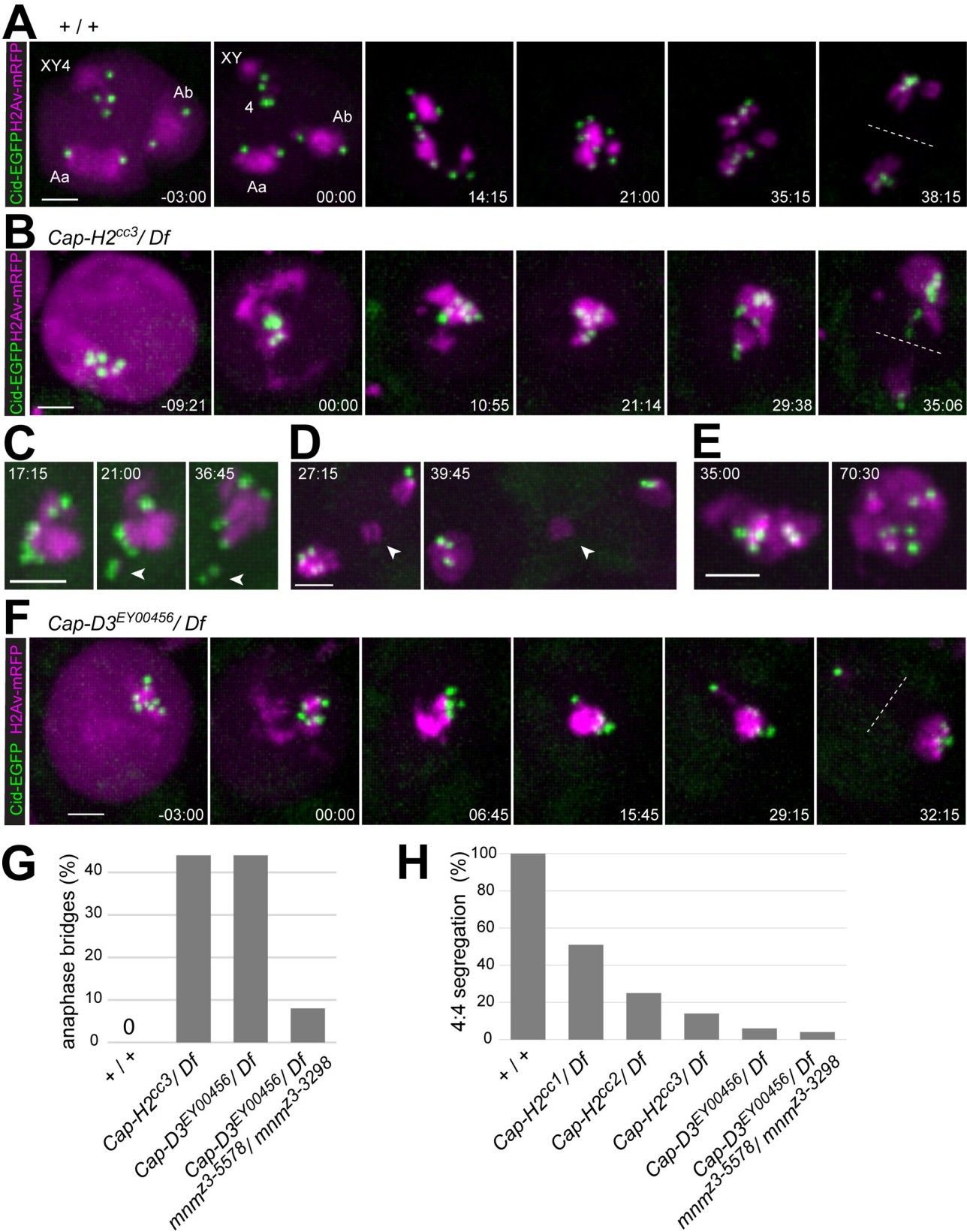

**Fig 4. Centromere and chromatin dynamics during M I in condensin II protein mutants.** Time-lapse imaging of His2Av-mRFP and Cenp-A/Cid-EGFP was applied for analysis of progression through M I in (**A**) control (+ / +), (**B-E**) *Cap-H2* mutants (*Cap-H2*$^{cc3}$/ *Df*), and (**F**) *Cap-D3* mutants (*Cap-D3*$^{EY}$/ *Df*). Time (min:sec) is indicated relative to the onset of NEBD I. (**C**) Arrowheads point to a chr4 bivalent that is separated out of the single clump of chromatin with clustered centromeres. (**D**) Arrowheads point to micronucleus resulting after bridge formation during anaphase. (**E**) Single clump of chromatin with clustered centromeres perduring into late anaphase and telophase after a complete failure of chromosome segregation. (**G**) Bars indicate the percentage of spermatocytes with chromosome bridges during anaphase I in the indicated genotypes. n = 48 from 5 cysts (+ / +), 62 from 10 cysts (*Cap-H2*$^{cc3}$/ *Df*), 48 from 6 cysts (*Cap-D3*$^{EY00456}$/ *Df*) and 28 from 6 cysts (*Cap-D3*$^{EY00456}$/ *Df*; *mnm*$^{z3-5578}$/ *mnm*$^{z3-3298}$). (**H**) Bars indicate the percentage of spermatocytes with 4:4 segregation of centromeres based on Cid-EGFP dot tracking during exit from M I. n = 48 from 5 cysts (+ / +), 45 from 8 cysts (*Cap-H2*$^{cc1}$/ *Df*), 8 from 2 cysts (*Cap-H2*$^{cc2}$/ *Df*), 63 from 10 cysts (*Cap-H2*$^{cc3}$/ *Df*), 46 from 6 cysts (*Cap-H*$^{EY00456}$/ *Df*) and 28 from 6 distinct cysts (*Cap-D3*$^{EY00456}$/ *Df*; *mnm*$^{z3-5578}$/ *mnm*$^{z3-3298}$). Scale bars = 3 μm.

*Exel6159* mutants, 4:4 segregation during anaphase was detected in only 14% of the cells (Fig 4H). Accordingly, 86% or more of the mutant spermatocytes failed at regular bi-orientation of bivalent centromeres. The *Cap-H2*$^{cc1}$ allele caused fewer mistakes in centromere bi-orientation (Fig 4H). In case of *Cap-H2*$^{cc2}$, which was less extensively analyzed, bivalent mis-orientation appeared to be intermediate (Fig 4H).

For comparison, we performed analogous time-lapse analyses with *Cap-D3*$^{EY00456}$/ *Df(2L) Exel7023* mutants. The abnormal phenotype observed during progression through M I in these mutants (Fig 4F) was highly similar to that in *Cap-H2*$^{cc3}$/ *Df(3R)Exel6159*. Chromatin bridges were displayed during anaphase I also in 44% of the spermatocytes and the fraction of cells with 4:4 segregation of centromeres during anaphase I was only 6% (Fig 4G and 4H).

In conclusion, our results confirm that the condensin II proteins Cap-D3 and Cap-H2 are essential for chromosome territory formation and regular chromosome segregation during M I [23]. Moreover, they provide additional insights into the role of chromosome territory formation. Previous studies have led to the proposal that chromosome missegregation during M I in *Cap-D3* and *Cap-H2* mutants arises from heterologous chromosome entanglements that exert their damaging effect during anaphase I by inhibiting chromosome separation [23]. Our analysis of centromere behavior revealed that meiotic chromosome segregation in *Cap-D3* and *Cap-H2* mutants is irregular for an additional reason. The failure of chromocenter break-up in the mutants has highly detrimental effects already before anaphase I. It generates a cluster of mechanically coupled centromeres that precludes regular bi-orientation of homologous centromeres during prometaphase I.

We add that our time-lapse imaging also revealed the unexpected finding that chromosome territories do not only form in spermatocytes but also in cyst cells in a condensin II-dependent manner (S6 Fig).

## Limited contribution of alternative homolog conjunction to meiotic chromosome missegregation in condensin II protein mutants

In principle, the chromosome bridges during exit from M I in *Cap-D3* and *Cap-H2* mutants, might result from a failure in elimination of alternative homolog conjunction (AHC). Normally, AHC is eliminated by separase-mediated cleavage of the AHC protein UNO during the metaphase-to-anaphase I transition [53]. To address whether condensin II proteins are required to remove AHC during M I, we performed time-lapse imaging with *Cap-D3*$^{EY00456}$/ *Df(2L)Exel7023* spermatocytes expressing UNO-EGFP.

Normally, UNO-EGFP is co-localized with other AHC proteins throughout spermatocyte maturation [53]. At the start of M I, it most strongly enriched on the chrXY pairing region, where it forms a very prominent dot that disappears within minutes at the onset of anaphase I (Fig 5A and 5C) [53]. UNO-EGFP signals on autosomes, which disappear concomitantly, are far lower and difficult to detect above background [53]. These weak autosomal UNO-EGFP signals are below detection limit with the chosen display settings (Fig 5A).

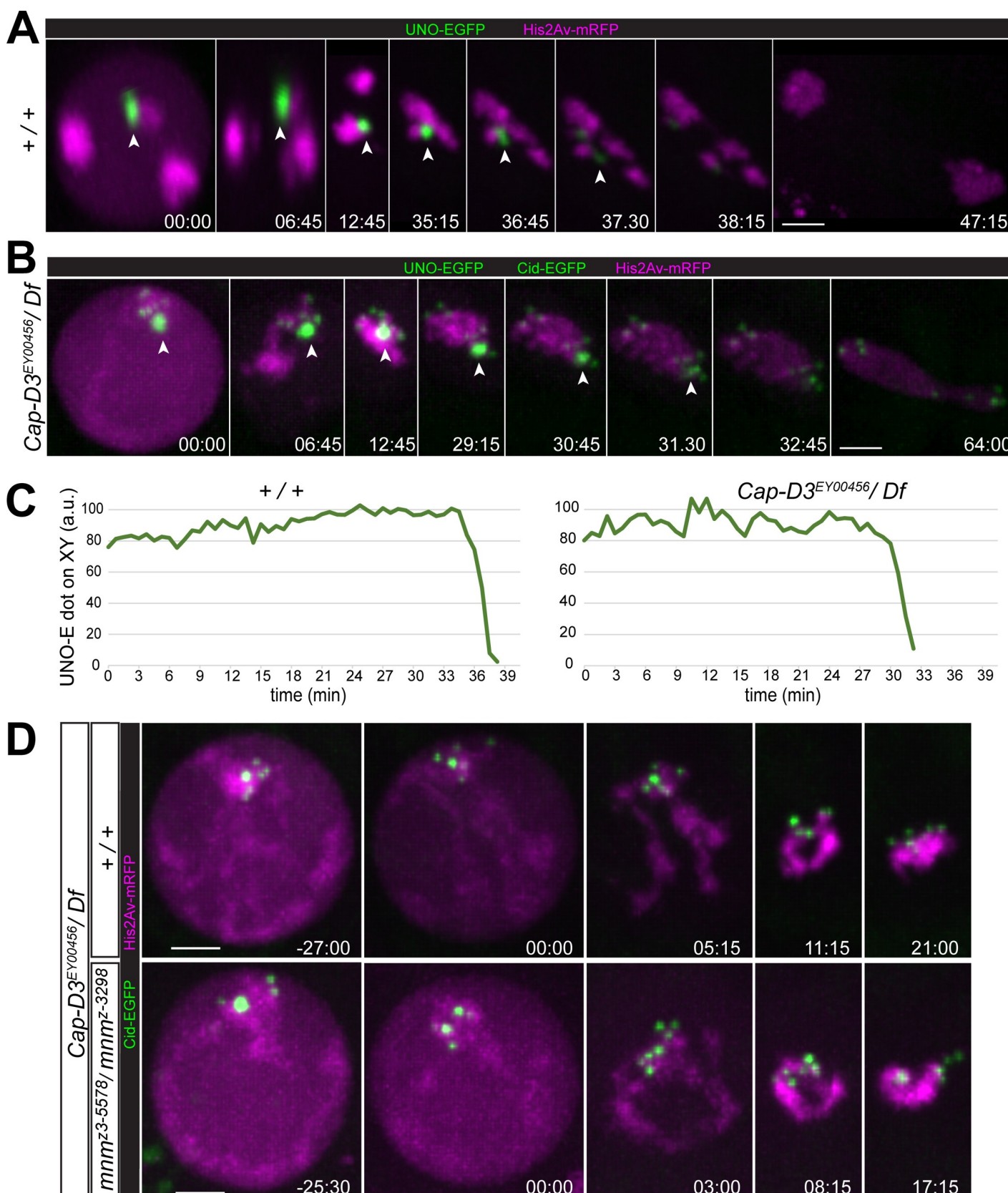

**Fig 5. Limited contribution of alternative homolog conjunction to meiotic chromosome missegregation in condensin II protein mutants.** (**A-C**) The disappearance of the alternative homolog conjunction protein UNO during M I was analyzed by time-lapse imaging in (**A**) control (+ / +) and (**B**) *Cap-D3* mutants (*Cap-D3*^EY00456/ *Df*). Beyond UNO-EGFP, spermatocytes expressed His2Av-mRFP, and in case of the *Cap-D3* mutants also Cenp-A/Cid-EGFP. The prominent UNO-EGFP dot on the chrXY pairing site (arrowheads) disappeared during anaphase I with comparable rapid kinetics in both genotypes, as confirmed (**C**) by quantification of the UNO-EGFP dot intensity over time. (**D**) Progression into and through M I was analyzed by time-lapse imaging in *Cap-D3* mutants (*Cap-D3*^EY00456/ *Df*) (top) and *Cap-D3 mnm* double mutants (*Cap-D3*^EY00456/ *Df; mnm*^z3-5578/ *mnm*^z3-3298) (bottom). Chromosomes condense into a single clump irrespective of presence or absence of AHC. Time (min:sec) indicated relative to the onset of NEBD I. Scale bars = 3 μm.

To detect the metaphase-to-anaphase I transition more clearly in our time-lapse imaging with *Cap-D3* mutant spermatocytes, they expressed not just UNO-EGFP and His2Av-mRFP but also Cenp-A/Cid-EGFP (Fig 5B). As shown previously [53], the strong UNO-EGFP dot on the chrXY pairing region can be distinguished readily from centromeric Cid-EGFP dots based on intensity and shape. As in normal spermatocytes, a single prominent UNO-EGFP dot was present in *Cap-D3* mutant spermatocytes just prior to NEBD I (Fig 5B), indicating that AHC protein localization is not affected, at least on the sex chromosome bivalent. However, we were unable to resolve whether autosomal UNO-EGFP dots are present or absent in *Cap-D3* mutant spermatocytes. Strong enhancement of the EGFP display settings, which reveals the weak autosomal UNO-EGFP dots in normal spermatocytes [53], did not allow a reliable identification of green signals with an unequivocal association with autosomal chromatin. Importantly, the highly prominent chromosomal UNO-EGFP dot that was present in *Cap-D3* mutant spermatocytes at the start of M I, disappeared rapidly around the metaphase-to-anaphase I transition, with kinetics indistinguishable from that in normal spermatocytes (Fig 5B and 5C). In conclusion, Cap-D3 is not required for the elimination of AHC during M I.

Although AHC is eliminated on time during exit from M I in *Cap-D3* mutants, it might still contribute to the meiotic chromosome segregation defects observed in these mutants, as previously suggested [23]. In condensin II protein mutants, conjunction might be established not just between homologous chromosomes but also between non-homologous chromosomes as a consequence of the failure of chromosome territory formation. Such ectopic AHC between non-homologous chromosomes might preclude timely disentangling of bivalents during spermatocyte maturation; the temporal window for disentangling remaining after AHC elimination in M I might be too short for a proper anaphase I without chromosome bridges. Addressing potential ectopic AHC by direct localization of AHC proteins is problematic, as barely detectable levels of AHC proteins are sufficient for stable conjunction of autosomal homologs during normal meiosis [52,53]. However, this issue has previously been addressed by phenotypic analysis of *Cap-H2 teflon* double mutants [23]. In contrast to loss of *teflon*, mutations in *mnm* results in a complete loss of AHC [52]. Therefore, we studied *Cap-D3 mnm* double mutants (*Cap-D3*^EY00456/ *Df(2L)Exel7023; mnm*^z3-5578/ *mnm*^z3-3298) by time-lapse imaging to resolve whether a complete AHC loss might result in an equally complete suppression of anaphase bridging (Fig 5D). Compared to *Cap-D3* single mutants, the percentage of spermatocytes with chromatin bridges during anaphase I was reduced in the *Cap-D3 mnm* double mutants, from 44% to 8% (Fig 4G). The fraction of cells with anaphase bridges remaining in the *Cap-D3 mnm* double mutants was still higher than in *mnm* single mutants, where anaphase bridge frequency is around 2% [72]. The incomplete suppression of bridges in *Cap-D3 mnm* double mutants indicates that some type of linkages other than those mediated directly by AHC contribute to the failure of chromosome segregation in condensin II protein mutants.

Interestingly, while anaphase bridging was substantially suppressed in *Cap-D3 mnm* double mutants compared to *Cap-D3* single mutants, centromere mis-orientation during M I was not. The fraction of cells with 4:4 centromere segregation, i.e., the upper bound for correct segregation, was comparably low in both *Cap-D3 mnm* double and *Cap-D3* single mutants (Fig 4H).

In fact, time-lapse imaging revealed very similar abnormalities at the start of M I in the double mutants as in the single mutants (Fig 5D). NEBD I was accompanied by formation of a single chromatin clump with clustered centromeres in both *Cap-D3* single and *Cap-D3 mnm* double mutants (Fig 5D). In contrast, in *mnm* single mutants (*mnm*[z3-5578]/ *mnm*[z3-3298]), bivalents were separated prematurely into univalents, as previously reported [52,55]. Moreover, our time-lapse analysis of centromere behavior during anaphase I in *mnm* mutants revealed a distribution of spermatocytes (n = 44) with either 4:4 (25.0%), 5:3 (43.2%), 6:2 (25.0%), 7:1 (6.8%) and 8:0 (0%) segregation, in excellent agreement with the mathematical prediction for the case of random segregation.

In summary, the importance of chromosome territory formation for regular chromosome segregation during M I goes beyond an avoidance of ectopic AHC between non-homologous chromosomes. Even in complete absence of AHC, non-homologous chromosome associations remain so prominent in *Cap-D3* mutants that a single clump of chromatin with clustered centromeres unfit for bi-orientation is generated at the start of M I.

## A TALE light system for analysis of chromocenter disruption dynamics during chromosome territory formation

While centromere clusters were disrupted in condensin II mutant spermatocytes just before NEBD I, the individualized centromeres nevertheless remained in close vicinity, apparently anchored by an intact chromocenter that also bundled the distinct emanating arm regions together (Fig 5D). The failure of chromocenter disruption in the mutants appeared to be largely responsible for the missegregation of chromosomes during M I. For specific analyses of chromocenter disruption, we adapted TALE lights for use in spermatocytes. TALE lights [73,74] are fusions of a fluorescent protein with a TALE domain, an engineered sequence-specific DNA binding domain. The DNA target sequences of the chosen TALE lights corresponded to those of two distinct satellites. A first red-fluorescent TALE light targeted the 359-bp satellite in the centromere-proximal heterochromatin of chrX (Fig 6A). A second green-fluorescent TALE light targeted the two 1.686 satellite blocks, which bind Prod [71] and are present in pericentromeric heterochromatin of chr2 and chr3 (Fig 6A). For use in spermatocytes, we generated *Drosophila* lines with *UASt* transgenes, designated as *UASt-RedX* and *UASt-Green2/3*, respectively. For validation, these transgenes were expressed in testes using *bamP-GAL4-VP16*. Immuno-FISH with antibodies recognizing the TALE lights and FISH probes for the targeted satellite loci confirmed that the TALE lights localized to their target sites (S7 Fig). Moreover, direct observation of TALE lights fluorescence revealed intranuclear dot signals in spermatocytes that were clearly above background (S7 Fig). The dot signals generated by *bam>RedX* were relatively weak and no longer detectable after the S4 stage. When detectable, there was a single RedX dot per nucleus, which was localized within the XY4 territory (S7 Fig). The dot signals resulting with *bam>Green2/3* were stronger and still detectable in early postmeiotic spermatids. The number of Green2/3 dots per nucleus was variable. One or two dots were present in early spermatocytes, and usually four dots in late spermatocytes, i.e., one dot pair per major autosome territory (S7 Fig). These dot numbers observed in spermatocytes were in agreement with previous FISH analyses [11]. In addition to the dot signals, *bam>RedX* and *bam>Green2/3* spermatocytes also displayed weaker diffuse nuclear signals.

For time-lapse imaging, *UASt-RedX* and *UASt-Green2/3* were co-expressed in spermatocytes with *bamP-GAL4-VP16*. Staging of spermatocytes was based on the nuclear diameter (d) revealed by the diffuse nuclear signals. Early spermatocytes with d < 9 μm, i.e., during S1 and S2 [9,10], displayed a single RedX dot (Fig 6B), as in the analyses with fixed samples. The number of Green2/3 dots was variable in early spermatocytes (Fig 6B), also as in fixed cells. In early

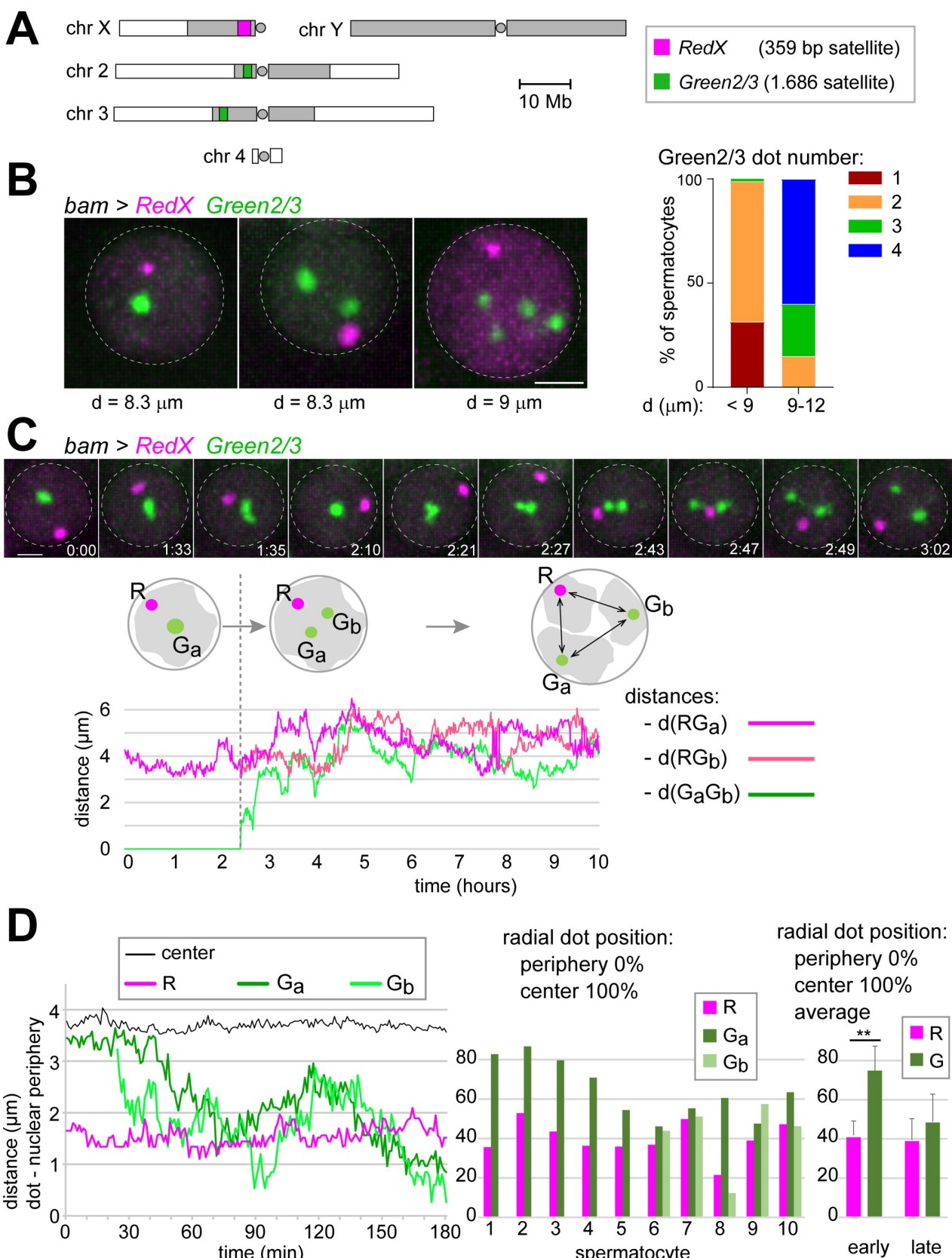

**Fig 6. Time-lapse imaging of chromocenter disruption in spermatocytes.** (**A**) The fluorescent TALE light proteins RedX and Green2/3 were used for analysis of chromocenter disruption by time-lapse imaging. The karyogram indicates the location of the 359 bp satellite locus (red) and the 1.686 satellite loci (green) that recruit RedX and Green2/3, respectively. Heterochromatic regions represented by grey shading and centromeres by circles. (**B**) Spermatocytes expressing RedX and Green2/3 (*bam>RedX Green2/3*) were analyzed by live imaging. The diameter (d) of the cell nucleus (dashed circle), as revealed by the weak diffuse red and green signals, was used as a proxy for developmental stage [9,10]. While a single RedX dot was present invariably during the stages S2 and S3, the number of Green2/3 dots was variable. Bar diagram for comparison of the number of Green2/3 dots in spermatocytes before (d < 9 μm) and after (d = 9–12 μm) chromosome territory formation. n = 270 (d < 9 μm) and 20 (d = 9–12 μm). (**C**) Tracking of RedX and Green2/3 dots over time. Still frames (top) illustrate the splitting of a Green2/3 dot into two widely separated dots, indicating chromocenter disruption. Time (h:min) indicated relative to the start of imaging. The distances separating RedX and Green2/3 dots in this spermatocyte were plotted over time (bottom). (**D**) Radial positions of RedX and Green2/3 dots during chromocenter disruption. Radial positions over time (left) after dot tracking in a representative spermatocyte. Bar diagram (middle) displays radial positions in five cells with a single Green2/3 dot (cells 1–5) and in five cells with two Green2/3 dots (cells 6–10). Bar diagram (right) provides the mean (± s.d.) after averaging the radial dot positions observed in cells 1–5 and 6–10, respectively. ** indicates p < .01 (t test). Scale bars = 2 μm.

spermatocytes (d < 9 μm), the number of Green2/3 dots per nucleus was one, two or three in 31%, 67% and 1% of the cells, respectively (Fig 6B). Therefore, in almost a third of the early spermatocytes, the 1.686 satellite blocks of chr2 and chr3 were in intimate association, as indicated by only a single Green2/3 dot per nucleus. This non-homologous association of pericentromeric heterochromatin regions of chr2 and chr3 reflects chromocenter formation. However, the 359-bp satellite block on chrX was not recruited into the region occupied by the 1.686 satellite loci of chr2 and chr3. Substantial overlap of Green2/3 and RedX dot signals was never observed. Nucleolus formation might separate the 359-bp satellite away from the main chromocenter because the 359-bp satellite is close to the rDNA locus of chrX.

Importantly, at later stages, in S3/S4 spermatocytes (d = 9–12 μm), non-homologous associations between the 1.686 satellite blocks of chr2 and chr3 were no longer apparent. None of the analyzed spermatocytes nuclei contained a single Green2/3 dot (Fig 6B). The majority (60%) displayed four Green2/3 dots, indicating that not just non-homologous associations were disrupted at this stage but also tight homolog pairing. These observations are in agreement with the conclusions of Vazquez et al. [10] that homologs are separated during stage S3 immediately after chromosome territory formation during stage S2b.

To analyze the temporal dynamics of chromocenter disruption, we tracked the RedX and Green2/3 dot signals in early spermatocytes (d = 7.1–8.3 μm). Spermatocytes with a single Green2/3 dot that was split into two during the imaging period were analyzed in detail (n = 20 from eight distinct cysts). Distances between the dots were determined, as well as their separation from the nuclear periphery (Fig 6C). After splitting of the Green2/3 dot, the distance between the two progeny dots increased up to 7.8 μm within one to two hours, with a dot speed around 0.03 μm/s. The single RedX dot was always relatively close to the nuclear periphery at the start of the chromocenter disruption phase and it remained peripheral over time (Fig 6D). In contrast, the single Green2/3 dot had a central position and the two progeny dots moved towards the nuclear periphery after splitting (Fig 6D).

In conclusion, fluorescent dot tracking in *bam>RedX* and *Green2/3* spermatocytes revealed the dynamics of chromocenter disruption.

## Stretching of 1.686 chromatin instead of efficient disruption of non-homologous 1.686 associations in condensin II protein mutants

For characterization of the consequences of the loss of condensin II proteins for chromocenter dynamics in spermatocytes, we analyzed *Cap-D3* and *Cap-H2* mutant spermatocytes with *bam>RedX Green2/3*. Beyond the presumed null alleles *Cap-D3*^EY00456^ and *Cap-H2*^cc3^, the hypomorphic mutation *Cap-H2*^cc1^ was analyzed as well. As in controls, the earliest spermatocytes (d < 9 μm, S1-S2) invariably displayed a single RedX and one or two Green2/3 dot signals in all three hemizygous mutant genotypes (Fig 7A). However, early spermatocytes with

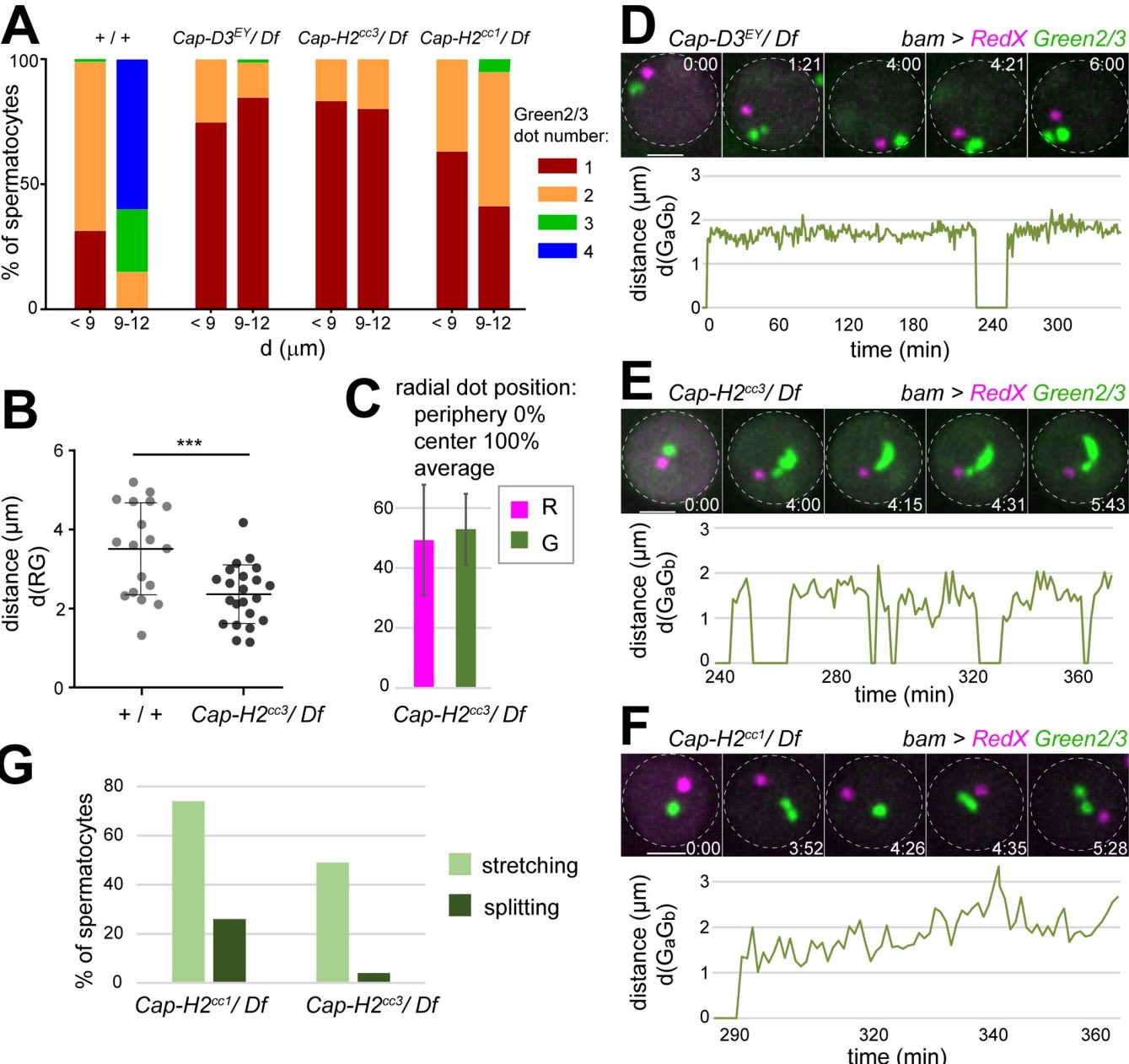

**Fig 7. Prolonged stretching of 1.686 satellite chromatin in the absence of condensin II proteins.** Spermatocytes with *bam>RedX Green2/3* in different genetic backgrounds were analyzed by live imaging. (**A**) The diameter (d) of the cell nucleus was used as a proxy for developmental stage. Bars indicate the number of Green2/3 dots in spermatocytes before (d < 9 μm) and after (d = 9–12 μm) chromosome territory formation in the indicated genotypes. Control data (+/+) is identical to that in Fig 6B. n = 270, 20, 119, 72, 155, 71, 212, 58 (from left to right). (**B**) Significantly shorter average separation between RedX and Green2/3 dots in *Cap-H2*[cc3] mutant spermatocytes compared to controls. Spermatocytes with one RedX and one Green2/3 dot at the S1/2 stage were analyzed. Mean ± s.d. is shown; n = 18 (+/+) and 22 (*Cap-H2*[cc3]/*Df*). (**C**) Average radial positions of RedX and Green2/3 dots (± s.d.; n = 6) in *Cap-H2*[cc3] mutant spermatocytes are not significantly different, in contrast to controls (Fig 6D). (**D-F**) Instead of the normal rapid splitting of single Green2/3 dots into two, condensin II protein mutants display episodes with prolonged Green2/3 dot stretching and occasional splitting. Time (h:min) is given relative to the start of displayed image sequence. Graphs display the extent of stretching over time. Spot detection (using IMARIS) resulted in the identification of two green spots after sufficient stretching even if a connection was still detectable in between. A distance value of zero indicates that only a single Green2/3 spot was detected. The plots start just before the stretching/splitting of the Green2/3 dot. Scale bars = 3 μm. (**G**) Percentage of S1-S4 spermatocytes that displayed at least one episode of Green2/3 dot stretching (as in panel E) or Green2/3 dot splitting (as in panel F) within the first 2.5 hours of imaging. n = 126 from 14 cysts (*Cap-H2*[cc1]/*Df*) and 45 from 4 cysts (*Cap-H2*[cc3]/*Df*).

only one Green2/3 dot were more than twofold more frequent in the mutants (Fig 7A). This increase was less pronounced with the hypomorphic *Cap-H2* allele compared to the strong *Cap-H2* and *Cap-D3* alleles (Fig 7A). In the early *Cap-H2*cc3 mutant spermatocytes with a single RedX and a single Green2/3 dot, their separation was significantly reduced compared to controls (Fig 7B) and the single Green2/3 dot was more peripheral (Fig 7C).

During spermatocyte maturation, differences between controls and condensin II protein mutants increased dramatically. In case of the strong alleles, the fraction of spermatocytes with non-homologous associations of chr2 and 3 (i.e., only a single Green2/3 dot) was largely maintained at around 80% (Fig 7A). In contrast, non-homologous associations were completely absent in controls at the S3/S4 stage (d = 9–12 μm) and even homolog pairing was already disrupted in the majority of cells (Fig 7A). In the hypomorphic *Cap-H2*cc1 hemizygotes, an increase of cells with two or even three Green2/3 dots at the expense of those with only one was evident when comparing S1/S2 with S3/S4 spermatocytes (Fig 7A). Thus, non-homologous associations were eliminated in *Cap-H2*cc1 mutants, although to a far lower extent than in controls.

Although chromocenter disruption did not succeed without condensin II proteins, the TALE lights time-lapse imaging exposed that some chromatin-separating forces were still at work in the mutants. In mutant spermatocytes with a single Green2/3 dot, a striking dynamic stretching of this signal occurred during the stages of chromosome territory formation and the subsequent stages S3 and S4 (Fig 7D–7G). Although transient stretching of the single Green2/3 dot was also observed occasionally in the controls just before definitive dot splitting, the stretching episodes lasted much longer in the mutants. To characterize the stretching episodes in the mutants, we applied software-based spot detection after time-lapse imaging (n = 10 spermatocytes for *Cap-D3*EY/*Df*, 23 for *Cap-H2*cc3/*Df*, and 21 for *Cap-H2*cc1/*Df*). With the chosen parameters, two spots were identified in the green channel not only after complete splitting of the Green2/3 dot but also after sufficient stretching of this dot. Once two spots were identified during a stretching episode, they were usually separated to about 2 μm (Fig 7D–7F). Occasionally separation reached up to 4.8 μm. Moreover, in a minority of the mutant spermatocytes, even a definitive splitting of the single Green2/3 dot into two could be observed. In about half of the analyzed *Cap-H2*cc3/*Df* spermatocytes during S2b-S4, we observed at least one Green2/3 dot stretching episode (Fig 7G). Definitive splitting was detected in only 2.2% of the *Cap-H2*cc3/*Df* spermatocytes (Fig 7G). Stretching episodes and definitive dot splitting were no longer detectable during S5 and S6. Compared to *Cap-H2*cc3, Green2/3 dot stretching, as well as its definitive splitting, were more frequent in the weaker *Cap-H2*cc1 allele (Fig 7G).

In conclusion, instead of the efficient disruption of non-homologous associations between the 1.686 satellite loci on chr2 and chr3 that accompanies normal territory formation, prolonged stretching of 1.686 chromatin rarely succeeding in definitive disruption of the non-homologous associations occurred in condensin II protein mutants. Thus, force generators other than condensin II appear to be at work during chromosome territory formation. However, in the absence of condensin II, they fail to achieve bivalent individualization. As documented (S8 Fig), we have addressed whether these unidentified force generators depend on cytoskeletal dynamics using inhibitors of F actin and microtubules. The inhibitors failed to affect both territory formation in control and Green2/3 dot stretching in condensin II protein mutants.

## The control of Cap-H2 expression levels in spermatocytes is crucial for bivalent formation

Analyses with *Drosophila* ovaries, larval salivary glands and cultured cells have demonstrated that Cap-H2 levels determine the extent of somatic homolog pairing, chromocenter formation

and centromere clustering [17,22,24,25,49]. In testis, the isoform F of Cap-H2 is most abundant according to RNA-Seq data [75]. Therefore, for an analysis of overexpression effects, we generated a line with an *UASt* transgene encoding this isoform with EGFP at the N terminus. *UASt-EGFP-Cap-H2* expression driven by *bamP-GAL4-VP16* was readily detected in early spermatocytes (Fig 8A). However, the EGFP-Cap-H2 signals were clearly above background only until about the S3 stage. In contrast, other EGFP fusion proteins, including those of the AHC proteins MNM, SNM and UNO [53,76], remain detectable throughout spermatocyte maturation when expressed analogously (i.e., using *bam> UASt*). Thus, EGFP-Cap-H2 appears to be a relatively unstable protein.

In early spermatocytes, EGFP-Cap-H2 was enriched on chromatin. Interestingly, the DNA staining pattern was abnormal in these spermatocytes. Already during the S1 stage, the DNA appeared to be more compact. At the S3 stage, more than the normal number of three territories were present (Fig 8B). The increased territory number was even more apparent at later stages, after nuclear growth and wider separation of the NE-associated territories (Fig 8B). Post-meiotic spermatids were also abnormal in *bam> EGFP-Cap-H2* testes. The size of spermatid nuclei was highly variable (Fig 8B), while normal testes display far less variation [53,76]. Finally, the fertility of *bam> EGFP-Cap-H2* males was reduced to about 40% of controls (Fig 8C).

For further characterization of the effects of EGFP-Cap-H2 overexpression on territories and meiotic chromosome segregation, we applied time-lapse imaging. First, we analyzed *bam> EGFP-Cap-H2* spermatocytes expressing also His2Av-mRFP and Cenp-A/Cid-EGFP. After NEBD I, instead of four normal bivalents, seven chromosomal entities were evident in *bam> EGFP-Cap-H2* spermatocytes (Fig 8D). Six of these had only one associated Cid-EGFP dot. These six displayed independent saltatory movements and failed to congress into the metaphase plate, indicating that they were univalents. The seventh chromosome with two associated Cid-EGFP dots displayed the asymmetries characteristic of the chrXY bivalent [55]. This apparently normal sex chromosome bivalent was regularly bi-oriented and segregated during anaphase I (Fig 8D). The same phenotype was observed consistently in all of the analyzed *bam> EGFP-Cap-H2* spermatocytes (n = 28, from four cysts).

To characterize the effects of *bam> EGFP-Cap-H2* in early spermatocytes, we analyzed the behavior of RedX and Green2/3 signals by time-lapse imaging (Fig 8E). The relatively weak nuclear EGFP-Cap-H2 signals in these spermatocytes during S1/2 did not preclude detection and tracking of the Green2/3 dots. In all the analyzed spermatocytes with *bam> EGFP-Cap-H2*, *RedX*, *Green2/3*, four completely resolved and independent Green2/3 dots were present already in S1/2 spermatocytes (n = 160) (Fig 8E). Tracking of these Green2/3 dots over time did not reveal permanent splitting, although some transient breathing of presumably sister chromatids was readily detectable. We conclude that increased levels of Cap-H2 appear to block the pairing of autosomal homologs, rather than inducing a premature separation of bivalents.

In conclusion, the effects of Cap-H2 overexpression emphasize the importance of control over Cap-H2 levels in early spermatocytes. Not only loss of Cap-H2 but also its excess is clearly detrimental for the success of male meiosis.

The importance of precise control of Cap-H2 levels in spermatocytes made the annotated D isoform interesting (Fig 3B). The D isoform of Cap-H2 is predicted to lack the N-terminal region required for binding to SMC2, whereas the conserved winged helix domain required for binding to SMC4 is present in the C-terminal region. While incapable of connecting the two core SMC subunits, the D isoform might act in an inhibitory manner by titrating Hawks away from condensin II and by blocking the binding of SMC4 to canonical Cap-H2 isoforms (if isoform D is still capable of binding to SMC4). The D isoform is also distinct from the

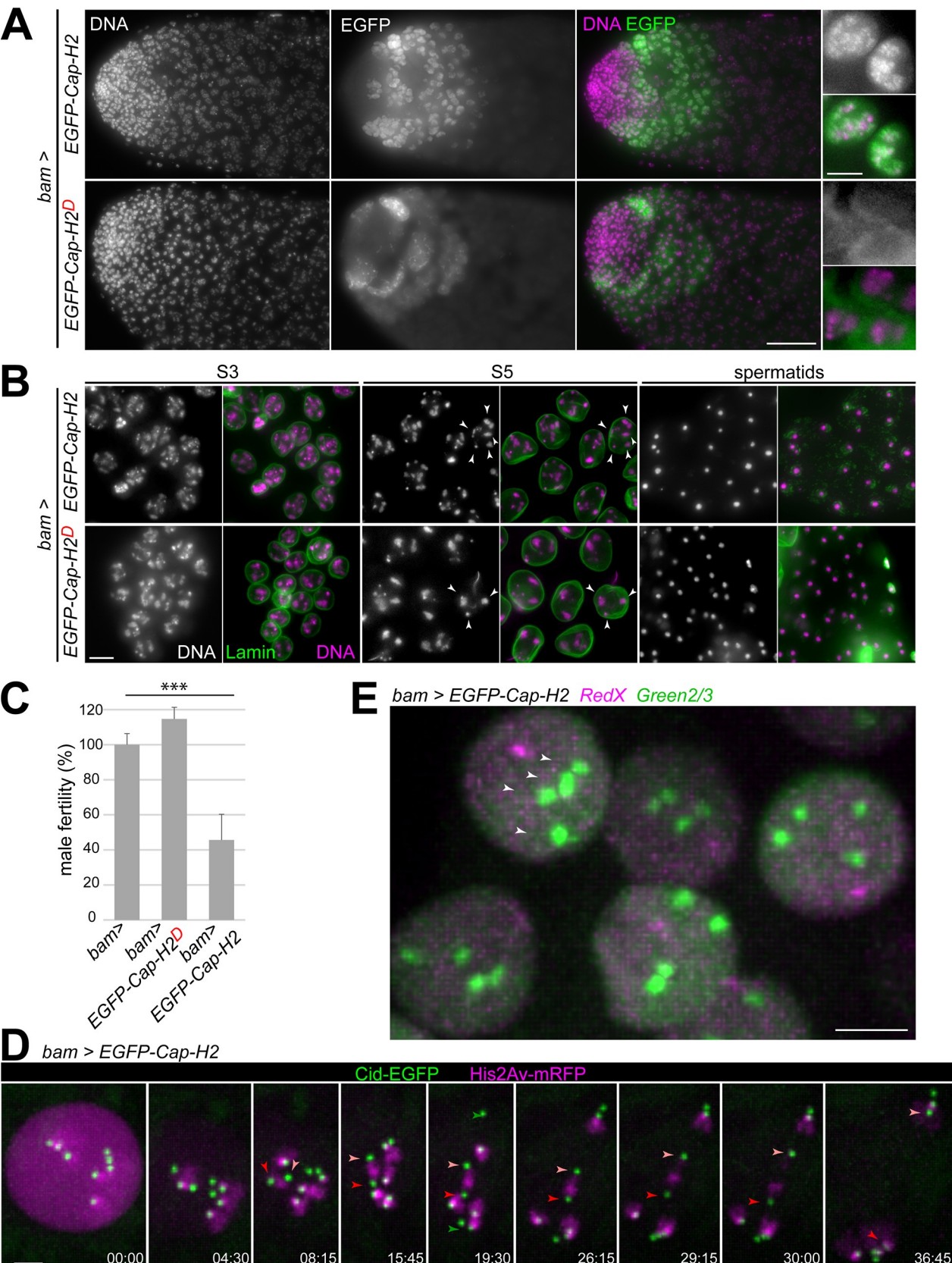

**Fig 8. Increased Cap-H2 levels inhibit the pairing of autosomal homologs.** (A-D) *UASt-EGFP-Cap-H2* (encoding isoform F) and *UASt-EGFP-Cap-H2$^D$* (encoding isoform D) were expressed in spermatocytes using *bamP-GAL4-VP16*. (A) EGFP signals and DNA staining in testis tip regions (left) and high magnification views with S3 spermatocytes (right). EGFP is present only transiently in early spermatocytes on chromatin (EGFP-Cap-H2) and cytoplasm (EGFP-Cap-H$^D$). (B) Expression of *UASt-EGFP-Cap-H2* results in too many chromosome territories (arrowheads) and meiotic chromosome missegregation in contrast to *UASt-EGFP-Cap-H2$^D$*. Spermatocytes at the stages S3 and S5 are displayed, as well as early round spermatids (sp), after labeling testis squash preparations with anti-Lamin Dm0 and a DNA stain. (C) Fertility of males with the indicated genotypes. Four test crosses, each with a single male parent, were set up and the resulting number of F1 progeny was counted. Mean fertility of *bam-GAL4-VP16* males without a UAS transgene (*bam>*) was set to 100%. *** indicates p < .001 (t test) and whiskers s.d. (D) Progression through MI was analyzed by time-lapse imaging of spermatocytes with *bam>EGFP-Cap-H2*, *His2Av-mRFP* and *Cenp-A/Cid-EGFP*. Time (min:sec) is indicated relative to the onset of NEBD I. Arrowheads indicate the centromeres of chr4 (green), chrX (dark red) and chrY (light red). After stable bi-orientation of the sex chromosome bivalent (t = 15:45), chrX and chrY are separated apart regularly during anaphase I (last three still frames). In contrast, autosomes are present as univalents at the start of M I, which segregate randomly after failure of congression into a metaphase plate (as indicated for example by chr4). (E) Live imaging of spermatocytes with *bam>EGFP-Cap-H2*, *RedX* and *Green2/3*. Four Green2/3 dots (arrowheads) are present already in early S1/2 spermatocytes. Scale bars = 40 μm (A, left), 10 μm (A, right), 5 μm (B, D), 3 μm (E).

canonical isoforms E–G in lacking a C terminal Slmb/β-TRCP binding site (Fig 3B), which keeps protein levels of canonical isoforms low [48]. Stabilization of the D isoform might boost its dominant-negative effect on condensin II activity. Moreover, this D isoform might be testis-specific according to RNA-Seq and EST data. Overall, it appeared conceivable, that the D isoform might down-regulate condensin II activity in spermatocytes. To evaluate the role of the D isoform, we generated lines with *UASt-EGFP-Cap-H2$^D$* integrated into the same chromosomal landing site as *UASt-EGFP-Cap-H2*. In *bam> EGFP-Cap-H2$^D$* spermatocyte, EGFP signals accumulated during the initial S1/2 stages predominantly in the cytoplasm with some enrichment on the fusome (Fig 8A). However, in contrast to expectations, EGFP-CapH2$^D$ was similarly unstable as EGFP-CapH2. EGFP signals were absent from late spermatocytes. Moreover, no obvious abnormalities with regard to chromosome territories were detected in *bam> EGFP-Cap-H2$^D$* spermatocytes (Fig 8B) and male fertility was normal (Fig 8C). In conclusion, these results do not support the speculation that the D isoform is a negative regulator of condensin II activity in spermatocytes. The function of the D isoform remains to be clarified.

## Discussion

Regular chromosome segregation during the first meiotic division does not just depend on prior pairing of homologs, but also on elimination of non-homologous chromosome associations. Perduring non-homologous associations preclude the regular bi-orientation and segregation of individualized bivalents during M I. In *Drosophila* spermatocytes, the process of chromosome territory formation was proposed to disrupt non-homologous associations [10], which are especially prominent between pericentromeric and centromeric chromosomal regions. Here, we have addressed potential mechanisms that promote the physical separation of bivalent chromosomes into distinct territories in *Drosophila* spermatocytes. Based on our time-lapse analyses of the spatial and temporal dynamics of centromeres and telomeres, in combination with experiments relying on F actin and microtubule inhibitors, we conclude that chromosome territory formation does not involve mechanisms analogous to those driving RPMs in diverse species, including yeast, *C. elegans* and mammals [3,4]. However, as previously shown [23], the condensin II proteins Cap-H2 and Cap-D3 are crucial for chromosome territory formation. Our detailed phenotypic characterizations with newly isolated mutants and novel fluorescent probes for live analysis of the dynamics of pericentromeric satellite loci demonstrate that the disruption of chromocenters, which is normally achieved within about 30–60 minutes during the S2b stage, does not succeed without Cap-H2 or Cap-D3. The failure of chromocenter disruption in *Cap-H2* and *Cap-D3* mutants results in chromosome condensation into a single clump around NEBD I, entirely incompatible with regular centromere orientation and segregation during M I. While condensin II is thus essential for territory formation,

we also found that this process appears to be driven by additional condensin II-independent forces.

In many species, RPMs of meiotic chromosomes are driven by motor proteins, which move on cytoplasmic F actin or MTs, dragging chromosomes along via LINC complex-mediated connections to distinct chromosomal regions. In *Drosophila* females, such machinery (LINC proteins, cytoplasmic dynein and MTs) promotes nuclear rotations primarily during interphase of the last gonial division cycle, fostering concomitant homolog pairing [56]. The possibility that analogous mechanisms might be exploited for chromosome territory formation in spermatocytes appeared likely, also because of the suggestion [10] that this process proceeds when each bivalent has a single centromere cluster, a potential convenient unique point of attack for the separating forces. Our analyses clearly refute this model. A state, where each of the four bivalents displays a single centromere cluster, is very rarely formed during the centromere de-clustering process that proceeds in early spermatocytes. Moreover, with separation distances often greater than 1.5 µm, a direct physical linkage of centromeres to LINC complexes appears unlikely. A similar wide separation from the nuclear periphery was observed for telomeres, which are LINC-bound for RPMs in mammals but apparently not in *Drosophila* spermatocytes. Finally, inhibitors of F actin and MTs (latrunculin B, cytochalasin D, colcemid) did not have noticeable effects on territory formation.

To study the role of condensin II proteins in chromosome territory formation in further detail, we have generated additional mutant *Cap-H2* alleles. The previously characterized alleles [23,24] were unlikely to affect all isoforms (*TH1*), hypomorphic and uncharacterized at the sequence level (*Z3-5163*) or associated with multiple SNPs including a relatively late premature stop codon (*Z3-0019*). The newly generated *cc3* allele has an early stop codon predicted to affect all annotated isoforms. In all likelihood, *cc3* eliminates gene function completely, although the N terminal Cap-H2 region that mediates binding to SMC2 might still be expressed. The alleles *cc1* and *cc2* carry in-frame deletions and are predicted to express β-kleisin variants that can still form a linkage between the heads of SMC2 and SMC4/Gluon, as both the corresponding N- and C-terminal binding regions are present. However, the central linker regions are truncated. Normally, this linker region contains the sequences for recruitment of the Hawk subunits, a Cap-D3- and a Cap-G2-like protein. These condensin II-specific Hawks were lost independently in various lineages [14,77,78]. While a *Cap-D3* is clearly present in the *Drosophila* genome, an obvious *Cap-G2* gene cannot be identified. Nevertheless, sequence conservation within the predicted Cap-G2-binding region of *Drosophila* Cap-H2 raises the possibility that a highly diverged *Drosophila* Cap-G2 might exist. Moreover, the notion that normal male meiosis in *Drosophila* requires a complete condensin II complex with Cap-H2 linking the two SMCs and recruiting both Hawks is consistent with the increasing severity of the phenotypes caused by the *cc1*, *cc2* and *cc3* mutations. The most severe phenotype was obtained with the *cc3* mutant, which can neither link the SMCs nor bind the Hawks. A slightly milder phenotype resulted with the *cc2* mutant, which might still link the two SMCs but fail to recruit the Hawks. An even milder phenotype was observed with the *cc1* mutant, which might not only link the two SMCs but also recruit Cap-D3. The *cc1* mutant lacks only the putative Cap-G2-binding region. Its abnormal phenotype might thus hint at a diverged and yet to be identified *Drosophila* Cap-G2. Clearly, however, alternative interpretations are not ruled out and additional detailed biochemical analyses will be required to clarify how Cap-D3 and Cap-H2 function in spermatocytes.

We point out that a condensin II complex was not detected by immunoprecipitation of EGFP-Cap-H2 or EGFP-SMC2 from ovary and embryo extracts followed by mass spectrometric characterization of co-precipitated proteins [79]. It is conceivable that this negative result reflects low abundance. Several lines of evidence, including RNA-Seq and proteome analyses

[80,81], indicate that *Cap-H2* expression levels are far lower than those of *barren*, which encodes the kleisin of condensin I. The canonical Cap-H2 isoforms E-G are also known to include a functionally important Slimb/β-TrCP degron that keeps protein levels low [48]. Low levels of Cap-H2 are important. Here, the overexpression of the canonical F isoform was shown to result in severe meiotic defects. As a caveat, we cannot exclude that the EGFP extension at the N-terminus of the F isoform that we have expressed in spermatocytes is causing the meiotic defects. However, this possibility seems unlikely, as overexpression of untagged Cap-H2 in other tissues [24,25] is known to have very similar effects as EGFP-Cap-H2 in spermatocytes. Accordingly, excess of Cap-H2 prevents the association of homologous chromosomal regions required for bivalent formation. Interestingly, in contrast to autosomal bivalents, the sex chromosome bivalent was not disrupted by Cap-H2 overexpression. In *Drosophila*, chrX and Y do not share any extended homology except for the presence of rDNA repeat loci, which mediate pairing during male meiosis [82]. AHC proteins accumulate to high levels on the paired rDNA loci, independent of *teflon* function [52]. In contrast, AHC proteins assemble on autosomal bivalents in far lower concentrations, dependent on *teflon* function [52]. Thus, AHC of autosomes and sex chromosomes is distinct mechanistically, presumably explaining the exclusive disruption of autosomal bivalents by Cap-H2 overexpression.

The effects of *Cap-H2* mutations or overexpression on interphase chromosome organization in spermatocytes correspond largely to those reported previously based on analyses in other species and other *Drosophila* cell types. In *Drosophila*, the polyploid nuclei of the larval salivary gland and of ovarian nurse cells, as well as cultured cell lines have been studied carefully [17,22–25,48,49]. A recent comparative study of interphase chromosome organization in 24 eukaryotic species from diverse evolutionary lineages using Hi-C revealed a striking correlation with the presence/absence of genes for condensin II-specific subunits [77]. Overall, there is an excellent consensus that condensin II controls interphase chromosome organization by axial compaction [25,49,77]. Without condensin II, chromosomes are less compacted lengthwise. Axial compaction by condensin II activity generally disrupts centromere clusters and chromocenters, as well as euchromatic homolog pairing in *Drosophila*, culminating in the establishment of chromosome territories with limited chromosome intermingling. In principle, the axial compaction of interphase chromosomes by condensin II might reflect DNA loop extrusion but other modes of action are not excluded. Importantly, chromosome territory formation in *Drosophila* spermatocytes provides a dramatic illustration that the global organization of chromosomes during interphase is not necessarily static. In fact, at the organismal level, carefully controlled condensin II activity appears to be of prime importance for male meiosis in *Drosophila*. Loss of Cap-D3 or Cap-H2 in an otherwise wild-type background has surprisingly mild consequences. It is fully compatible with development into morphologically normal adults. While mutant males are completely sterile, female fertility remains high even though chromosome organization in nurse cells is not normal in mutants [25]. The disruptive effects of condensin II activity on non-homologous associations between centromeres, pericentromeric heterochromatin and telomeres appear to be minimal in nurse cells [25] and maximal in spermatocytes, indicating cell-type specific modulation.

In *Drosophila*, interphase genome organization with centromere clustering, chromocenter formation and homolog pairing is observed almost throughout development and in most cell types. These chromosomal associations are all disrupted for chromosome individualization at the onset of mitosis even in the absence of condensin II proteins. However, in preparation for M I in spermatocytes, only the non-homologous associations of centromeres and pericentromeric heterochromatin need to be disrupted, whereas, importantly, homologous associations have to be preserved at least partially. Instead of COs, *Drosophila* spermatocytes employ AHC, which protects against premature bivalent disruption at the onset of M I. It remains to be

clarified how AHC is targeted correctly. Likely, AHC is applied after chromosome territory formation, resulting in appropriate linkages between homologs and avoidance of inappropriate non-homologous linkages. However, the establishment of AHC needs to occur early after chromosome territory formation [76], presumably before the forces, which drive this process, have also disrupted all homologous associations by collateral damage. Accordingly, it appeared likely that in the absence of condensin II proteins, AHC might be established ectopically also between non-homologous chromosomes, thereby explaining the formation of a single chromosome clump at M I onset. Our evidence indicates that AHC is indeed established in condensin II protein mutants, although we do not know whether also ectopically. Importantly, however, a single chromosome clump at M I onset is also formed unchanged in *Cap-D3 mnm* double mutants. Therefore, the chromosome clumping is not caused exclusively by putative ectopic AHC linkages. Why then is chromosome individualization before M I in spermatocytes so critically dependent on condensin II, in contrast to all other mitotic and meiotic divisions? The large size of the spermatocyte nucleus might matter. In condensin II mutant spermatocytes, the intermingled chromosome arms are dramatically extended all along the NE. The chromosome condensation and individualization machinery at work during onset of M phase (primarily condensin I and topoisomerase II) might not be effective enough to achieve sufficient individualization within the given time.

In condensin II protein mutant spermatocytes, AHC is not only established but also removed at the metaphase-to-anaphase transition of M I according to our time-lapse imaging with UNO-EGFP. The aberrant chromosome structure in these mutants might compromise AHC removal efficiency to some extent, explaining the frequent and often transient anaphase I bridges in these mutants and their reduction in *Cap-D3 mnm* double mutants (this work) and *Cap-H2 teflon* double mutants [23]. The mechanistic details how the opposing activities of condensin II and AHC are regulated and coordinated during spermatocyte maturation clearly deserve additional attention.

While condensin II is clearly of paramount importance for chromosome territory formation in spermatocytes, it is unlikely to provide all the forces required for the eventual wide spatial separation of bivalents. Axial chromosome compaction by dynamic DNA loop extrusion (in combination with topoisomerase II activity) is a highly attractive mechanism for disentangling bivalents, but it is more difficult to envisage how this might directly generate the substantial, apparently DNA-free gaps that are in between the bivalents in mature spermatocytes. Much of this intervening nuclear space is taken up by the Y chromosome loops, which spread out during spermatocyte maturation in parallel with the increasing distancing of bivalent territories [83,84]. However, Y loops are unlikely to drive territory separation, as territories appear to be normal in X0 spermatocytes. Another potential contribution to territory separation might arise from nuclear envelope growth. In wild-type spermatocytes, chromosome territories are intimately associated with the NE. This association in combination with the massive NE expansion during spermatocyte growth has been proposed as a putative mechanism for territory separation [10]. As shown here, this mechanism is definitely not responsible for the initial territory separation, which occurs rapidly without concomitant nuclear growth at a stage, where DNA staining does not reveal an intimate association with the nuclear lamina.

Our RedX and Green2/3 imaging provided interesting hints at condensin II-independent forces that appear to contribute to territory formation. The dynamic stretching and occasional splitting of the Green2/3 dot observed in condensin II protein mutants suggests that attempts at chromocenter disruption continue in these mutants. Temporally these attempts appear to be confined primarily to the stages S2 to S4, i.e., when territories are formed initially and later when homologs and sisters are largely separated according to studies with lacO/lacI-GFP [10]. Additional analyses will be required to confirm this temporal confinement. Moreover, it

should be of great interest to identify the molecular basis of the forces that are responsible for the Green2/3 dot stretching in these mutants. Our analyses with inhibitors argue against an involvement of microtubules and F actin, which has been implicated in intranuclear movements of chromosomal loci in *Drosophila* [85]. Condensin I, cohesin and SMC5/6 complexes might be additional candidates deserving exploration. Recent analyses after spermatocyte-specific depletion of condensin I proteins, which appear to have a predominant cytoplasmic localization also in spermatocytes, have failed to expose an involvement of this SMC complex in chromosome territory formation, although insufficient depletion remains as alternative explanation [86].

In conclusion, our work has led to novel tools and insight concerning the mechanisms of chromosome territory formation, as also summarized in S9 Fig. However, it is obvious that our understanding of this fascinating dynamic and global rearrangement of interphase chromosomes remains speculative and much needs to be learned for a mechanistic explanation at the molecular level.

## Materials and methods

### *Drosophila* lines

The lines with the following mutations or transgenes have been described before: *Df(3R) Exel6159* (Bloomington *Drosophila* stock Center (BDSC) # 7638), *Df(2L)Exel7023* (BDSC # 7797), *Cap-D3*[EY00456] (BDSC # 7797), *mnm*[Z3-3298] and *mnm*[Z3-5578] [52], *cav*[EGFP] (EGFP-HOAP) [61], *g-uno-EGFP III.1* [53], *His2Av-mRFP* and *gCid-EGFP-Cid* [87], *UbiP{GFP (S65T)-βTub56D}17–1* (DGRC Kyoto # 109603), *P{y[+t7.7] w[+mC] = D1-GFP.FPTB}attP40* (BDSC # 66454), *bamP-GAL4-VP16* [88].

Lines with the following transgenes were generated by microinjection (BestGene Inc., Chino Hills, CA, USA) with the plasmid constructs described further below: *UASt-EGFP-Cap-H2*, *UASt-EGFP-Cap-H2*[D], *UASt-RedX*, and *UASt-Green2/3*. The first two pUASt-attB constructs were integrated into the landing site *P{CaryP}attP40*. The *UASt-RedX* transgene in the established *Drosophila* lines was observed to be unstable. Initial analyses of testis from *bam> RedX* males clearly resulted in a single nuclear red dot in early spermatocytes. Subsequent analogous analyses after standard maintenance of the *UASt-RedX* lines for about 3 years in the laboratory did not reveal these red dots as previously in the initial experiments. Depending on the subline, dots were either substantially weaker or no longer detectable above the diffuse red nuclear signal that was still present. The *UASt-RedX* coding sequence encompasses 20 TALE repeats of 102 bp, which are identical in sequence except for 2–6 central base pairs in some repeats. Therefore, recombination between repeats might occur occasionally and result in altered protein products with weakened or abolished DNA binding activity. Such instabilities were not encountered in case of *UASt-Green2/3* even though it contains the same number of TALE repeats.

The *Cap-H2* alleles (*Cap-H2*[cc1], *Cap-H2*[cc2], and *Cap-H2*[cc3]) were generated using CRISPR/ Cas9. Embryos from *y*[1] *M{w[+mC] = Act5C-Cas9.P}ZH-2A w** (Bloomington *Drosophila* stock Center #54590) were injected (BestGene Inc., Chino Hills, CA, USA) with pCFD5 derived plasmids described further below. Each of these pCFD5 derivatives allowed expression of a gRNA pair targeting two distinct gene regions. Adults obtained from the injected eggs were crossed singly with *w**;; *Sb/TM3, Ser*. A fraction of the F1 progeny was used for preparation of genomic DNA and analysis by PCR using the primers LV048 and LV049 (see S1 Table for sequences of all synthetic DNA oligonucleotides). These primers annealed 337 bp upstream and 490 bp downstream of the two gRNA targets sites, respectively. If the PCR analysis indicated the presence of CRISPR/Cas9-induced *Cap-H2* deletion alleles among the tested progeny

of a particular founder animal, additional F1 progeny males derived from the same founder were crossed singly with *w*;; Sb/TM3*, *Ser* to establish balanced lines. These lines were tested again by PCR with LV048 and LV049 for the presence of *Cap-H2* deletion alleles. Moreover, the amplified fragments were sequenced. The sequences around the intragenic deletion breakpoints (-II-) were: 5'-AGCGAAGTCGAG–II–CCCACATTTGAC-3') (*Cap-H2*[cc1]), 5'-CGACAA GCGCTTCAACGC–II–TCGACCCACATTTG-3' (*Cap-H2*[cc2]) and 5'-ATGTCGGACGACAAGC GCTTCAACG–II–GGCTTTGTAAAGT-3' (*Cap-H2*[cc3]). In case of *Cap-H2*[cc3], a second intragenic deletion, which is identical to that in *Cap-H2*[cc1], was detected further downstream. The chromosomes carrying *Cap-H2*[cc1] and *Cap-H2*[cc3] are homozygous lethal but hemizygous viable, suggesting the presence of second-site recessive lethal mutations on these chromosomes. The chromosome carrying *Cap-H2*[cc2] is homozygous viable but male sterile.

*X0* males were generated by crossing virgin females isolated from the stock *C(1;Y)2, y B/0 & C(1)RM, y v/0)* (BDSC #2487) with X/Y males (w[1118]).

Standard crossing was used for the generation of the various strains used for experimental analyses. The genotypes of the flies analyzed are described in detail in the supporting information (S2 Table). All flies analyzed were raised at 25°C.

## Plasmids

For generation pCFD5-Cap-H2_gRNAs1-2, a synthetic double stranded DNA fragment (LV046) (Integrated DNA Technologies, Coralville, IA, USA) was inserted into of pCFD5 [89] after digestion of vector and insert with BbsI.

For generation of pUASt-RedX, a modified version of pUASt was generated first. After digestion of pUASt with EcoRI and XbaI, a linker generated by annealing oligonucleotides CL305 and CL306 was inserted. This linker insertion resulted in the elimination of the EcoRI and XbaI sites and in the introduction of a NheI and an XhoI site. pET28-359_2-mCherry [73] was digested with NheI and XhoI to release the region coding for 359 TALE Light-mCherry, which was inserted into the corresponding newly introduced restriction sites of the modified pUASt vector.

For the generation of pUASt-Green2/3, we modified pUASt-mcs-EGFP [90] in a first step by digestion with EcoRI and XhoI, followed by insertion of a linker generated by annealing the oligonucleotides CL318 and CL319. This linker insertion resulted in the introduction of a NheI and an XhoI site. pET28_CM3_EGFP [73] was digested with NheI and SalI to release the 1.686-TALE coding region, which was inserted into the newly introduced restriction sites of the modified pUASt-mcs-EGFP vector.

For the generation of our UASt-Cap-H2 transgene constructs, we modified pUAS-p1-EGFP-Cap-H2 [79] (kindly provided by S. Heidmann, Unversity of Bayreuth). This starting plasmid had been constructed with a cDNA isolated from an EST clone (SD18322), which contained the coding sequence for the isoform G with a few genetic polymorphisms. This coding region was mutated into a version that codes for the isoform F. Mutagenesis was performed as described [91] with the primer CL316. The region coding for EGFP-Cap-H2 (isoform F) was then isolated after digestion with NotI and XhoI and transferred into the corresponding sites of a pUASt-attB variant. This variant was obtained by digestion of pUASTattB [92] with EcoRI and BglII, followed by insertion of a linker generated by annealing oligonucleotides CL312 and CL313. With a final step, the *Cap-H2* 3'UTR sequences were deleted by digestion with EcoRI and XhoI, followed by insertion of a linker obtained by annealing the oligonucleotides CL314 and CL315. The final construct pUASt-attB-EGFP-Cap-H2 (isoform F) contains the SV40 3'UTR sequences, as present in pUASTattB, downstream of the EGFP-Cap-H2 coding sequence. For generation of pUASt-attB-EGFP-Cap-H2 (isoform D), we first

deleted the sequences coding for the isoform F-specific N-terminal region from pUASt-EGFP-Cap-H2 (isoform F) as described [91] using the primer CL317. A NotI-AatII fragment was isolated from the mutagenized plasmid and used for replacement of the NotI-AatII fragment in pUASt-attB-EGFP-Cap-H2 (isoform F) resulting in the last intermediate plasmid. In a final step, we converted the C-terminal coding sequences to those present in isoform D. This was achieved with a synthetic DNA fragment (CL320) (GenScript Biotech, Leiden, Netherlands) of about 550 bp flanked by AatII and EcoRI restriction sites. This synthetic DNA fragment was used for replacement of the AatII-EcoRI fragment in the last intermediate, yielding the final construct pUASt-attB-EGFP-Cap-H2 (isoform D). The presence of the correct insert sequences in all of our constructs was verified by DNA sequencing.

## Fertility tests

For analysis of male fertility, we crossed single males with three *w* virgin females. Five to ten replicate crosses were started. For analysis of female fertility, three virgin females of the genotype to be tested were pooled and crossed with three *w* males. Ten replicate crosses were set up. After two days of mating, crosses were transferred into a fresh vial. After an additional two days, all adult flies were discarded, followed by counting all of the adult progeny that developed subsequently at 25˚C.

## Fixation and labeling of testis preparations

For whole mount testis preparations, dissection was performed in testis buffer (183 mM KCl, 47 mM NaCl, 10 mM Tris-HCl, pH 6.8). Testes were fixed in phosphate buffered saline (PBS) containing 4% formaldehyde and 0.1% Triton X-100 in 0.2 ml Eppendorf tubes for 20 minutes on a rotating wheel. For immunolabeling, mouse monoclonal anti-Lamin Dm0 antibody ADL67.10 (Developmental Studies Hybridoma Bank, University of Iowa, Iowa City, IA, USA) was used at 1:50 and the rabbit antibody against Prod [71] at 1:5000. Secondary antibodies were Alexa568- or Alexa488-conjugated goat antibodies against mouse IgG (Invitrogen, A11004 and A10631) or Alexa568-conjugated goat antibodies against rabbit IgG (Invitrogen, A11011) diluted 1:500. For DNA staining, testes were incubated for 10 minutes in PBS, 0.1% Triton X-100 (PBTx) containing Hoechst 33258 (1 μg/ml). After three washes with PBS, testes were transferred into a drop of mounting medium (70% glycerol, 1% n-propyl gallate, 0.05% p-phenylenediamine, 50 mM Tris-HCl pH 8.5) on a slide before adding a cover slip.

Testis squash preparations were made and stained essentially as described previously [93], according to protocol 3.3.2, except that the mounting medium described above was used. For immunolabeling, mouse monoclonal anti-Lamin Dm0 antibody ADL67.10 (Developmental Studies Hybridoma Bank) was used also at 1:50 and rabbit anti-Cid IS1 [94] was used 1:1000. Secondary antibodies were used as described above.

For immuno-FISH, testes were dissected and fixed with 4% formaldehyde in PBS, followed by permeabilization with PBS containing 0.3% Triton X-100 and 0.3% sodium deoxycholate. Immunolabeling was done as described above with the following antibodies: rabbit anti-EGFP (IS28) [90] diluted 1:3000 or rabbit anti-mCherry (IS751) diluted 1:3000. Alexa488-conjugated goat anti-rabbit IgG (Invitrogen, A11008) diluted 1:1000 was used as secondary antibody. Ethanol incubations and dehydration with a formamide series were done as described (immuno-FISH protocol 3.2, steps 10–26) [95]. An oligonucleotide (5'-TTTTCCAAATTTCGGTCAT-CAAATAATCAT-3') with Atto-565 on both the 5'- and the 3' end (Integrated DNA Technologies, Leuven, Belgium) was used for detection of the X-specific 359 bp satellite sequences at a concentration of 1 ng/μl in hybridization buffer. An oligonucleotide (5'- ATAACATAGAA-TAACATAGAATAACATAGA -3') with Atto-565 on both the 5'- and the 3' end (Integrated

DNA Technologies, Leuven, Belgium) was used for detection of the 1.686 satellite sequences in pericentric heterochromatin of chr2 and chr3 at a concentration of 1 ng/μl in hybridization buffer. The denaturation step was performed at 98°C for 6 min, and hybridization over night at 18°C. Slides were washed twice for ten minutes for each wash with 50% formamide, 2x SSCT at 18°C. Thereafter, additional washes of ten minutes were performed at room temperature, first once in 25% formamide, 2x SSCT and then three times in 2x SSCT. DNA was stained with Hoechst 33258 (1 μg/ml) for 10 minutes. Before mounting, slides were washed twice in PBS for 5 minutes.

Generally, about 20 dissected testes were mounted per slide. Images (z stacks) were acquired using a Zeiss Cell Observer HS wide-field microscope using 40×/0.75 or 63×/1.4 objectives, except for the images presented in S2 Fig. These latter images were acquired with an Olympus FluoView 1000 laser-scanning confocal microscope using a 60×/1.42 objective.

## Testis preparations for live imaging

Time-lapse imaging of progression through meiosis was performed as recently described [55]. In brief, testes from pupal or young adult males were dissected in Schneider's *Drosophila* Medium (Invitrogen, #21720) supplemented with 10% fetal bovine serum (Invitrogen) and 1% penicillin/streptomycin (Invitrogen, #15140). The dissected testes were transferred into 45 μl of medium in a 35 mm glass bottom dish (MatTek Corporation, #P35G-1.5-14-C) and opened with fine tungsten needles to release the cysts. To reduce sample movements, 15 μl of 1% w/v methylcellulose (Sigma, #M0387) was added. A wet filter paper was placed inside along the dish wall before sealing the lid with parafilm. For long-term time-lapse imaging of territory formation over up to six hours, 1.5 ml of medium and 0.5 ml of methylcellulose were used. No wetted filter papers were used in these experiments.

Drugs were administered by pipetting small volumes of stock solution directly to the final testis preparation in the 35 mm dishes. Colcemid (demecolcine, Sigma, #D6165) was dissolved in DMSO (10 mM) before further dilution in tissue culture medium and used at a final concentration of 10 μM. Latrunculin B (Sigma-Aldrich L5288) was also dissolved in DMSO (10 mM) before further dilution in tissue culture medium and used at a final concentration of 4 μM. Cytochalasin D (Sigma-Aldrich C2618) was further diluted in DMSO (1 mM) and used at a final concentration of 10 μM.

Imaging was performed at 25°C in a room with temperature control using a spinning disc confocal microscope (VisiScope with a Yokogawa CSU-X1 unit combined with an Olympus IX83 inverted stand and a Photometrics evolve EM 512 EMCCD camera, equipped for red/green dual channel fluorescence observation; Visitron systems, Puchheim, Germany). A 60×/1.42 oil immersion objective was used for acquisition of z stacks. The z stacks acquired for analysis of progression through M I comprised 46 focal planes spaced by 500 nm and the stacks were acquired at 45 seconds intervals (Figs 2A and 2B, 4, 5, S6, S8 panel A and B). The z stacks acquired for the analysis of chromosome territory formation comprised 29 focal planes spaced by 800 nm. The time intervals between stack acquisitions were 10 minutes (Fig 1A–1D), one minute (Figs 6, 7, S8 panel C—E), 30 seconds (Fig 1E), 10 seconds (Fig 2), or five seconds (Figs 1F and S3).

## Image processing and analysis

Maximum intensity projections were generated using ImageJ for wide-field images and IMARIS (Bitplane; versions 8.4.0, 9.2.0, 9.7.2) for confocal images. For measuring the distance of Cid-EGFP dots from the nuclear lamina, an isosurface was created based on the anti-Lamin Dm0 signal using IMARIS for image segmentation. The same software was used for spot

detection based on the Cid-EGFP signals, setting the parameter "estimated xy diameter" to 500 nm with background subtraction as described [55]. Distance values were obtained by using the function "distance transformation" of the IMARIS software. The resulting values were exported as a Microsoft Excel file.

For the analysis of chromosome segregation during M I, centromeric Cid-EGFP signals were tracked after spot detection as described above and the algorithm "Autoregressive Motion" of IMARIS software. The same procedure was also applied for the analysis of centromere behavior during chromosome territory formation. Telomeric EGFP-HOAP dots were tracked analogously using an estimated xy diameter of 400 nm. The software identified the majority of the centromeric Cid-EGFP and telomeric EGFP-HOAP dots correctly with the chosen parameter settings. Tracking of the spots over time resulted in tracks that contained at most a few incorrect linkages without significant effects on the results, if the image stacks sequences had been acquired at maximal spatial and temporal resolution (Figs 1F, 2D and 2E, and S3). For time-lapse data acquired at a lower frame rate, tracks were corrected as follows. Spots for centromeric or telomeric signals that were not recognized automatically were added manually. Similarly, spots assigned to background signals were deleted. Manual correction of tracks was readily possible because the distinct features of the His2Av-mRFP signals associated with centromeric or telomeric dot signals could also be taken into account, while these were ignored during the automatic detection by the IMARIS software. Centromeric and telomeric signals observed during progression through M I were assigned to specific chromosomes using criteria as previously described [55]. Moreover, NEBD and later transitions during progression through M I were also scored as described [55].

For the quantification of centromeric Cid-EGFP signal intensities as well as UNO-EGFP signals associated with the XY pairing center (Figs 2B and 2E and 5C), IMARIS software was used for image segmentation. The parameters used for the generation of isosurfaces were iteratively modified until signals of interest were completely surrounded by the isosurface. Signal intensities of voxels within a given isosurface were summed and exported as a Microsoft Excel file.

To measure the spatial separation of centromeric or telomeric dot signals from the nuclear periphery (Figs 1E and 1F and 2D), we used an analogous procedure as described above for images acquired with Cid-EGFP and anti-Lamin Dm0 signals (S2 Fig). However, instead of using the anti-Lamin Dm0 signal the His2Av-mRFP signal was used for the generation of an isosurface around the nuclear periphery. The resulting separation distances were exported as a Microsoft Excel file.

To measure the velocity of the movements of tracked centromeric and telomeric dot signals using IMARIS software (Figs 1G, 2E, and S3), we also exploited the His2Av-mRFP signals for the generation of an isosurface around the periphery of the nucleus of interest. After tracking the isosurface over time, its translational drift was corrected with the IMARIS software. The velocity of the residual movements of the tracked centromeric and telomeric dot signals at each time point were then exported as a Microsoft Excel file.

RedX and Green2/3 dots signals were tracked during the stages of chromosome territory formation using the spot detection function of IMARIS. In this case, the parameter "estimated xy diameter" was set to 1 μm. To determine radial positions of these dots, an isosurface was created around the nucleus based on the weak diffuse nucleoplasmic RedX signals. Thereafter, the separation of the dots from the nuclear periphery was determined as described above. The weak diffuse nucleoplasmic RedX signal was also used to determine the radius of the nucleus. The distances of each dot from the nuclear periphery at the first 10 time points of imaging were exported as a Microsoft Excel file, which was used for calculation of the radial positions and their mean. To measure the distance between two dots (Figs 6C and 7E and 7F; d(RG$_a$), d

(RG$_b$) and d(G$_a$G$_b$)), spot positions at each time point of the track were exported as a Microsoft Excel file. Python scripts were used for further analysis and calculation of the separation distances.

Figures display maximum intensity projections unless stated otherwise. The projections were generated with either ImageJ or IMARIS. Export of projections from IMARIS as movies or still frames after live imaging was made with interpolated image display. Moreover, display parameters for the His2Av-mRFP signals were adjusted manually over time to reveal chromosomes clearly throughout the movies, thereby correcting photobleaching and partially also the changes in the extent of chromosome condensation during M I. Graphs were generated with Microsoft Excel or GraphPad Prism. P values were calculated using a two tailed student t test (* = $p < 0.05$; ** = $p < 0.01$; *** = $p < 0.001$). Adobe Photoshop, Adobe Illustrator and Inkscape were used for production of figures.

## Supporting information

**S1 Fig. Variable dynamics of centromere de-clustering during chromosome territory formation.** Spermatocytes expressing His2Av-mRFP and Cenp-A/Cid-EGFP were analyzed by time-lapse imaging. The number of centromeric Cid-EGFP dots was determined and plotted over time. The graphs from a subset of the analyzed spermatocytes are displayed for illustration of the considerable temporal variability of centromere de-clustering. t = 0 was set at the transition from the three- to the four-dot stage.
(PDF)

**S2 Fig. Spatial separation of centromeres from the nuclear lamina.** (**A**) Whole mount testis preparations were labeled with anti-Lamin Dm0 and a DNA stain. Maximum intensity projections with representative spermatocytes at the indicated stages illustrate that the large majority of Cid-EGFP dots is not intimately associated with the nuclear lamina. In case of the S1/2 spermatocyte with three Cid-EGFP dots (top), optical sections cutting through these dots are presented below the projection. (**B**) Dot plot presenting the separation distance between Cid-EGFP dots and the nuclear lamina at the indicated stages. The number of the analyzed dots is given in the plot, as well as the mean distance (± s.d.). Number of spermatocytes analyzed = 6 (S1/2), 9 (S3), 4 (S4), and 5 (S5). Scale bars = 1 μm (S1/2) and 3 μm (S3 –S5).
(PDF)

**S3 Fig. Centromere mobility.** Testes expressing Cenp-A/Cid-EGFP and His2Av-mRFP were used for time-lapse imaging at five-seconds intervals. Three to five cells (c1-c5) at the indicated stages were analyzed. Each Cid-EGFP dot in these cells was tracked over time. Dots were numbered, except for the stages S1 and S2, where designations as in Fig 1A were applied. All velocity values observed for a given Cid-EGFP dot during an eight min period were plotted (n = 99), as well as the mean ± s.d.
(PDF)

**S4 Fig. *Cap-H2*$^{cc}$ mutations abolish DNA partitioning into territories during spermatocyte maturation.** (**A,B**) Squash preparations of testes with the indicated genotypes were stained for DNA. Spermatocytes at the indicated stages are displayed. In control spermatocytes (+ / +), the three chromosome territories containing either large autosomes (Aa and Ab) or the other chromosomes (XY4) are evident at the S4 and even more clearly at the S5 stage (arrowheads). In contrast, territories are absent in *Cap-H2* and *Cap-D3* mutant spermatocytes. Scale bars = 5 μm.
(PDF)

**S5 Fig. Behavior of heterochromatin proteins D1 and Prod in *Cap-H2*[cc3] mutant spermatocytes.** (**A, B**) Whole mount preparations of testes expressing D1-sfGFP were labeled with anti-Prod, anti-Lamin Dm0 and a DNA stain. Testes were either from control males (+ / +) of from *Cap-H2*[cc3] / *Df(3R)Exel6159* mutants. (**A**) Testis tip regions. (**B**) High magnification view with spermatogonial cells (upper region) and S1/2 spermatocytes (lower region) with high and low levels of D1-sfGFP, respectively. Scale bars = 20 μm (A) and 10 μm (B).
(PDF)

**S6 Fig. Condensin II proteins are required for chromocenter and centromere cluster dissociation in somatic cyst cells.** (**A—C**) Time-lapse imaging of His2Av-mRFP and Cenp-A/Cid-EGFP was applied for analysis of cyst cells in late spermatocyte cysts in (**A**) control (+ / +), (**B**) *Cap-D3* mutants (*Cap-D3*[EY] / *Df*), and (**C**) *Cap-H2* mutants (*Cap-H2*[cc3] / *Df*). Time (min:sec) is indicated relative to the onset of imaging. Scale bars = 3 μm. See S6 Fig for further explanations.
(PDF)

**S7 Fig. A transgenic TALE light system for time-lapse imaging of chromocenter disruption during territory formation in spermatocytes.** Squash preparations of testes from males with *bamP-GAL4-VP16* and (**A**) either *UASt-RedX* (*bam>RedX*) or *UASt-Green2/3* (*bam>Green2/3*), or (**B**) both together (*bam>RedX Green2/3*) were labeled with a DNA stain. (**A**) Immuno-FISH labeling. FISH probes targeting either the 359 bp satellite or the 1.686 satellite sequence were used in combination with either anti-mCherry for detection of RedX (arrowheads) or anti-EGFP for detection of Green2/3. High magnification views with S3 spermatocytes are displayed. (**B**) High magnification views with spermatocytes at the indicated stages are displayed. The three chromosome territories formed by the large autosomes chr2 and chr3 (Aa and Ab) and by the other chromosomes (XY4) are indicated on the left. Scale bars = 5 μm.
(PDF)

**S8 Fig. Inhibitors of F actin and microtubules do not interfere with chromosome territory formation in controls and with Green2/3 dot stretching in condensin II protein mutants.** See S8 Fig for further explanations.
(PDF)

**S9 Fig. Chromosome territory formation in *Drosophila* spermatocytes.** Scheme summarizing implicated mechanisms including explanations.
(PDF)

**S1 Movie. Chromosome territory formation and centromere de-clustering.** Spermatocytes expressing Cenp-A/Cid-EGFP (green) and His2Av-mRFP (magenta) were analyzed by time-lapse imaging at 10 min intervals. The progression from a stage with three Cid-EGFP dots to a stage with six such dots and clearly separated domains of chromatin-associated His2Av-mRFP, as observed in the spermatocyte presented in Fig 1A, is shown in a maximum intensity projection.
(MP4)

**S2 Movie. Number and spatial positions of telomeric EGFP HOAP dots during progression through M I.** Spermatocytes expressing EGFP-HOAP and His2Av-mRFP were analyzed by time-lapse imaging at 45 sec intervals. Progression through M I, as observed in the spermatocyte presented in Fig 2A, is shown in a maximum intensity projection. The image sequence is repeated three times. In the first and second period, EGFP-HOAP (green) and His2Av-mRFP signals (magenta) are shown without and thereafter with dots marking telomeric signals. During the third period, only HOAP-EGFP signals are shown as grey values.

Time (h:min:sec) is indicated with t = 0 at start of the movie.
(MP4)

**S3 Movie. Progression through M I in control spermatocyte.** Spermatocytes expressing Cenp-A/Cid-EGFP and His2Av-mRFP were analyzed by time-lapse imaging at 45 sec intervals. Progression through M I, as observed in the spermatocyte presented in Fig 4A, is shown in a maximum intensity projection. Time (min:sec) is indicated with t = 0 at start of the movie.
(MP4)

**S4 Movie. Progression through M I in *Cap-H2*<sup>cc3</sup>/ *Df* spermatocyte.** *Cap-H2*<sup>cc3</sup>/ *Df* spermatocytes expressing Cenp-A/Cid-EGFP and His2Av-mRFP were analyzed by time-lapse imaging at 45 sec intervals. Progression through M I, as observed in the spermatocyte presented in Fig 4B, is shown in a maximum intensity projection. A lateral drift correction after tracking the strongest Cid-EGFP centromere signal was applied to keep the spermatocyte at the center of the movie until onset of anaphase I. Time (min:sec) is indicated with t = 0 at start of the movie.
(MP4)

**S5 Movie. Progression through M I in *Cap-D3*<sup>EY00456</sup>/ *Df* spermatocyte.** *Cap-D3*<sup>EY00456</sup>/ *Df* spermatocytes expressing Cenp-A/Cid-EGFP and His2Av-mRFP were analyzed by time-lapse imaging at 45 sec intervals. Progression through M I, as observed in the spermatocyte presented in Fig 4F, is shown in a maximum intensity projection. Time (min:sec) is indicated with t = 0 at start of the movie.
(MP4)

**S6 Movie. Disappearance of UNO-EGFP during M I in *Cap-D3*<sup>EY00456</sup>/ *Df* spermatocyte.** *Cap-D3*<sup>EY00456</sup>/ *Df* spermatocytes expressing UNO-EGFP, Cenp-A/Cid-EGFP and His2Av-mRFP were analyzed by time-lapse imaging at 45 sec intervals. Progression through M I, as observed in the spermatocyte presented in Fig 5B, is shown in a maximum intensity projection. The prominent UNO-EGFP dot on the sex chromosome pairing region is marked by a yellow circle and the weaker centromeric Cid-EGFP dots by small white dots and tracks during exit from M I. A lateral drift correction after tracking the prominent UNO-EGFP signal was applied to keep the spermatocyte at the center of the movie until onset of anaphase I. Time (min:sec) is indicated with t = 0 at start of the movie.
(MP4)

**S7 Movie. Chromosome condensation at M I onset in *Cap-D3*<sup>EY00456</sup>/ *Df* spermatocyte.** *Cap-D3*<sup>EY00456</sup>/ *Df* spermatocytes expressing Cenp-A/Cid-EGFP and His2Av-mRFP were analyzed by time-lapse imaging at about 45 sec intervals. Progression into M I, as observed in the spermatocyte presented in Fig 5D (top), is shown in a maximum intensity projection. Onset of NEBD I starts at 27:09 in the movie. To reveal the spatial arrangements of chromatin during the chromosome condensation process, progression from late S6 until metaphase I is interrupted in the movie by periodic nuclear rotations (360˚ horizontal followed by 360˚ vertical) at the time points 00:00, 27:09, 32:29 and 36:51. Time (min:sec) is indicated with t = 0 at start of the movie.
(MP4)

**S8 Movie. Chromocenter disruption in an early control spermatocytes revealed by *bam>UASt-RedX Green2/3*.** Spermatocytes with *bam>UASt-RedX Green2/3* were analyzed by time-lapse imaging at 60 sec intervals. The splitting of a Green2/3 dot into two, as observed in the spermatocyte presented in Fig 6C, is shown in a maximum intensity projection. Time

(min:sec) is indicated with t = 0 at start of the movie.
(MP4)

**S9 Movie. Green2/3 dot stretching in an early *Cap-H2*<sup>cc3</sup>/ *Df* spermatocyte.** *Cap-H2*<sup>cc3</sup>/ *Df* spermatocytes with *bam>UASt-RedX Green2/3* were analyzed by time-lapse imaging at 60 sec intervals. The stretching of a Green2/3 dot, as observed in the spermatocyte presented in Fig 6E, is shown in a maximum intensity projection. Time (h:min:sec) is indicated with t = 0 at start of the movie.
(MP4)

**S10 Movie. Green2/3 dot splitting in an early *Cap-H2*<sup>cc1</sup>/ *Df* spermatocyte.** *Cap-H2*<sup>cc1</sup>/ *Df* spermatocytes with *bam>UASt-RedX Green2/3* were analyzed by time-lapse imaging at 60 sec intervals. The splitting of a Green2/3 dot into two, as observed in the spermatocyte presented in Fig 6F, is shown in a maximum intensity projection. Time (h:min:sec) is indicated with t = 0 at start of the movie.
(MP4)

**S11 Movie. Progression through M I after *Cap-H2* overexpression.** Spermatocytes expressing *Cenp-A/Cid-EGFP* and *His2Av-mRFP*, as well as *UASt-Cap-H2* driven by *bamP--GAL4-VP16* were analyzed by time-lapse imaging at 45 sec intervals. Progression through M I, as observed in the spermatocyte presented in Fig 8D, is shown in a maximum intensity projection. Time (h:min:sec) is indicated with t = 0 at start of the movie.
(MP4)

**S1 Table. Oligonucleotide sequences.**
(XLSX)

**S2 Table. Description of the analyzed genotypes.**
(XLSX)

**S3 Table. Source data.**
(XLSX)

## Acknowledgments

We thank Kai Yuan, Pat O'Farrell and Stefan Heidmann for providing plasmids with the TALE light and Cap-H2 coding sequences, and Tibor Török for anti-Prod. Moreover, we are grateful to Joe Weber for the Python scripts and Sina Moser for technical support.

## Author Contributions

**Conceptualization:** Luisa Vernizzi, Christian F. Lehner.

**Data curation:** Luisa Vernizzi, Christian F. Lehner.

**Formal analysis:** Luisa Vernizzi, Christian F. Lehner.

**Funding acquisition:** Christian F. Lehner.

**Investigation:** Luisa Vernizzi, Christian F. Lehner.

**Methodology:** Luisa Vernizzi.

**Project administration:** Christian F. Lehner.

**Supervision:** Christian F. Lehner.

**Validation:** Luisa Vernizzi, Christian F. Lehner.

**Visualization:** Luisa Vernizzi, Christian F. Lehner.

**Writing – original draft:** Christian F. Lehner.

**Writing – review & editing:** Luisa Vernizzi, Christian F. Lehner.

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
