## [Decision Letter · Decision Letter 0]

13 Sep 2021

Dear Christian

Thank you very much for submitting your really very nice article entitled 'Bivalent individualization during chromosome territory formation in Drosophila spermatocytes by controlled condensin II protein activity and additional force generators' to PLOS Genetics.

The manuscript was fully evaluated at the editorial level and by independent peer reviewers. The reviewers appreciated the attention to an important topic but identified some concerns that we ask you address in a revised manuscript. While we ask that you consider all of the issues raised by the three reviewers, we ask that you pay specific attention to the concerns that the paper could be significantly shortened. We also ask that you address the point regarding 4:4 segregation events that was raised by Reviewer 3.

We therefore ask you to modify the manuscript according to the review recommendations. Your revisions should address the specific points made by each reviewer.

[LINK]

Yours sincerely,

R. Scott Hawley

Associate Editor

PLOS Genetics

Gregory P. Copenhaver

Editor-in-Chief

PLOS Genetics

Reviewer's Responses to Questions

**Comments to the Authors:**

Reviewer #1: This study by Vernizzi and Lehner studies Drosophila spermatocyte meiosis, where chromosome territory formation replaces the need of crossover between homologs. How the territory formation (i.e. separation of non-homologous chromosomes from each other) is achieved remains largely a mystery. This study utilized live observation combined with multiple fluorescently tagged markers to decipher the process. They focus on condensin subunits Cap-D3 and -H2, which have been known to be required for territory formation. They first describe the centromere and telomere behaviors in wild type in detail, then the comparison with condensin mutants revealed that these condensin subunits are required for centromeres declustering that is normally observed after the completion of chromosome territory formation. This explains why condensins are required for territory formation, during drosophila male meiosis, as described previously.

Overall, data quality is high, and the discoveries described in this manuscript contributes to our understanding of how homologous ‘pairing’ is achieved in drosophila male meiosis, which does not use canonical pairing mechanism, thus is of considerable interest. Although the mechanistic understanding is not there yet, their high quality cytology offers plenty of information that provides important insights into how territory may be formed, and how condensins may contribute to it.

I found the writing to be very long and descriptive, to the point that it interferes with the understanding of general readers. For example, figure titles should be conclusion, instead of ‘what the author did’ (for example, instead of ‘time-lapse analysis of telomere behaviors’, they should state ‘telomeres do not cluster in Drosophila spermatocytes’ etc.). Also, description is quite lengthy: I understand the importance of being descriptive to document important subtleties but it is also important that the paper can reach to broader readership. I recommend to make the manuscript a bit more concise and add clearer conclusions /take home messages more frequently throughout the manuscript.

Specific comments

Abstract is very reader-unfriendly. Chromocenter is not defined. Line 28-31 difficult to understand. ‘persistence of centromere clusters ….. in the mutants…. do not depend on those proteins’: which protein are they talking about?

Line 687-688: ‘Figure 22A’?

Reviewer #2: This is an important paper because it is a rigorous genetic and functional of condensin II in Drosophila. The problems with previous studies of condensin II have been that the null phenotype has not been d, since existing alleles were not lethal. In this paper, the authors report on a careful analysis of CRISPR induced Cap-H2 alleles, one of which is determined to be a null allele. They also conclude that an existing allele of Cap-D3 is a null allele. Based on these mutations, they find that condensin II is required for male meiosis and fertility but not for females or viability. In mutant male meiosis, there is a striking elimination of territory formation, centromere declustering and subsequent defects in chromosome segregation during meiosis I. One of the observed phenotypes is chromosome bridges at meiotic anaphase. This phenotype is not associated with a defect in removal of an AHC protein (UNO), but does partially depend on AHC protein MNM. Another strength of this paper are its technical accomplishments, including live imaging and the development of TALE lights to live image the behavior of centromeres.

1) My most significant criticism is the length of this paper. The Introduction and Discussion are very long and should be shortened to make the paper more accessible. The first results section (and two figures) are descriptive and only loosely connected to the rest of the paper. They do not involve the condensin mutants and the conclusions in line 397-398 are correlative and not based on direct testing of telomere function. The significance of Figure 2D and 2E is difficult to appreciate. In short, it seems the Results section could start with line 401 and figure 3.

2) The dependence of the bridging phenotype on mnm is interesting. It might make sense to present the mnm data before UNO. That is, the mnm data shown that most of the bridges depend on AHC proteins. However, this might be indirect because UNO localization is not affected. In the same section, Fig 4 should include comparisons to the mnm single mutant. I also don’t think the argument for “hypothetical ectopic AHC” is very strong. There is no evidence that the effect of mnm is due to ectopic AHC. Delete the discussion of Teflon (lines 625-631) because there is no data with it. Could move to discussion.

3) Most observations and data well documented. But a few cases where the conclusions could be backed up with more quantitation. This includes lines 540-541 and line 821. Was EGFP-Cap-H2 overexpression verified with a measurement? (line 827).

4) The section on stretching and splitting is confusing. It would benefit from a better definition of these two events, does it occur in wild-type, and its relationship or difference compared to territory formation. Stretching and splitting is introduced as something observed in the mutants (line 750) and Figure 7 lacks any measurement in wild-type. Where is the control data for Fig 7D-G. Is it in Fig 6C? Figure 8 has images of wild-type but the graph is for the mutant. Furthermore, my feeling is that the section from lines 765 – 806 adds little to the paper, especially if stretching and splitting is a mutant phenotype rather than a normal event. This paper has a few cases of negative results and this is probably the least informative of those.

5) Lines 545-548 is confusing and should be deleted. Especially since the frequency of 4:4 segregation was so low. The fact that some of the 4:4 could also be defective could be more simply stated.

6) Line 566: Is the additional reason for segregation errors a manifestation of the same defect in territory organization?

7) Line 344: “centromere s”

8) Lines 569 – 586 is a distraction.

9) Line 752: cc3/Df is written twice.

Reviewer #3: Please see attachment

**Have all data underlying the figures and results presented in the manuscript been provided?**

Reviewer #1: Yes

Reviewer #2: Yes

Reviewer #3: Yes

PLOS authors have the option to publish the peer review history of their article (what does this mean?). If published, this will include your full peer review and any attached files.

Reviewer #1: **Yes: **Yukiko Yamashita

Reviewer #2: No

Reviewer #3: **Yes: **Bruce D. McKee

---

## [Editor Report · Decision Letter 1]

11 Oct 2021

Dear Dr Lehner,

Thank you for submitting this truly lovely revision. It is such a gift when revisions are as thoughtful and thorough as this one is. So, yes, it is ACCEPTED!

We are pleased to inform you that your manuscript entitled "Bivalent individualization during chromosome territory formation in Drosophila spermatocytes by controlled condensin II protein activity and additional force generators" has been editorially accepted for publication in PLOS Genetics. Congratulations!

Yours sincerely,

R. Scott Hawley

Associate Editor

PLOS Genetics

Gregory P. Copenhaver

Editor-in-Chief

PLOS Genetics

Comments from the reviewers (if applicable):

**Data Deposition**

http://datadryad.org/submit?journalID=pgenetics&manu=PGENETICS-D-21-01044R1

**Press Queries**

---

## [Editor Report · Acceptance letter]

15 Oct 2021

PGENETICS-D-21-01044R1 

Bivalent individualization during chromosome territory formation in Drosophila spermatocytes by controlled condensin II protein activity and additional force generators 

Dear Dr Lehner, 

We are pleased to inform you that your manuscript entitled "Bivalent individualization during chromosome territory formation in Drosophila spermatocytes by controlled condensin II protein activity and additional force generators" has been formally accepted for publication in PLOS Genetics! Your manuscript is now with our production department and you will be notified of the publication date in due course.

With kind regards,

Zsofia Freund

PLOS Genetics

On behalf of:
